# The extra-islet pancreas supports autoimmunity in human type 1 diabetes

Graham L Barlow[1,2]*, Christian M Schürch[2,3], Salil S Bhate[2], Darci J Phillips[2], Arabella Young[4,5,6], Shen Dong[4,7], Hunter A Martinez[1], Gernot Kaber[1], Nadine Nagy[1], Sasvath Ramachandran[1], Janet Meng[1], Eva Korpos[8], Jeffrey A Bluestone[4,7,9]*, Garry P Nolan[2]*, Paul L Bollyky[1]*

[1]Division of Infectious Diseases and Geographic Medicine, Department of Medicine, Stanford University School of Medicine, Stanford, United States; [2]Department of Pathology, Stanford University School of Medicine, Stanford, United States; [3]Department of Pathology and Neuropathology, University Hospital and Comprehensive Cancer Center, Tübingen, Germany; [4]Diabetes Center, University of California, San Francisco, San Francisco, United States; [5]Huntsman Cancer Institute, University of Utah Health Sciences Center, Salt Lake City, United States; [6]Department of Pathology, University of Utah School of Medicine, Salt Lake City, United States; [7]Sean N. Parker Autoimmune Research Laboratory and Diabetes Center, University of California, San Francisco, San Francisco, United States; [8]Institute of Physiological Chemistry and Pathobiochemistry and Cells-in-Motion Interfaculty Center, University of Muenster, Muenster, Germany; [9]Sonoma Biotherapeutics, South San Francisco, United States

**\*For correspondence:**
grahaml_barlow@dfci.harvard.edu (GLB);
jbluestone@sonomabio.com (JAB);
gnolan@drowlab.com (GPN);
pbollyky@stanford.edu (PLB)

## eLife Assessment

This **valuable** study leverages innovative high-dimensional imaging strategies to interrogate pancreatic immune cell profiles and distributions throughout stages of type 1 diabetes (T1D). Despite a notable limitation in the number of donor samples analyzed, the authors identify a series of intriguing "immune signatures" and histopathological features that collectively constitute a **solid** foundation for future investigations into immunological processes underpinning the pathogenesis of T1D. Accordingly, the work will be of considerable interest to the community of T1D researchers and clinicians.

**Abstract** In autoimmune type 1 diabetes (T1D), immune cells infiltrate and destroy the islets of Langerhans — islands of endocrine tissue dispersed throughout the pancreas. However, the contribution of cellular programs outside islets to insulitis is unclear. Here, using CO-Detection by indEXing (CODEX) tissue imaging and cadaveric pancreas samples, we simultaneously examine islet and extra-islet inflammation in human T1D. We identify four sub-states of inflamed islets characterized by the activation profiles of CD8[+]T cells enriched in islets relative to the surrounding tissue. We further find that the extra-islet space of lobules with extensive islet-infiltration differs from the extra-islet space of less infiltrated areas within the same tissue section. Finally, we identify lymphoid structures away from islets enriched in CD45RA[+] T cells — a population also enriched in one of the inflamed islet sub-states. Together, these data help define the coordination between islets and the extra-islet pancreas in the pathogenesis of human T1D.

**eLife digest** When someone has type 1 diabetes, their immune system mistakenly targets and destroys β-cells in the pancreas, which produce insulin, the hormone that helps bring down sugar levels in the blood after we eat. Despite advances in treatment, most people with type 1 diabetes will depend on insulin for their entire lives.

T cells are a type of immune cell involved in type 1 diabetes. These cells infiltrate the pancreatic islets, the structures where β-cells reside, to attack the β-cells. This process, called insulitis, is poorly understood, partly because obtaining tissue samples containing islets in the process of being infiltrated by T cells is extremely difficult.

Barlow et al. collaborated with the Network for Pancreatic Organ Donors with Diabetes to obtain pancreatic tissues from eight organ donors with type 1 diabetes, and two organ donors whose immune systems could recognize islets but who were not yet exhibiting diabetes symptoms.

Barlow et al. analysed 54 proteins in each tissue section and examined how inflammation progressed in islet cells and the surrounding pancreas to better understand insulitis. The researchers identified four types of insulitis, each defined by the types of T cells present. The nature of the T cells in islets is important because it may affect how fast type 1 diabetes progresses. Although Barlow et al. did not examine enough cases to establish if a correlation exists between the types of insulitis and disease progression, this can be examined in future studies. They also found that, during insulitis, the blood vessels in the islets switched on a protein called IDO, possibly in response to T cells that infiltrate islets. IDO may temporarily protect the islets from the immune response as insulitis progresses, but it is insufficient to protect the β-cells. Barlow et al. further found aggregates of T cells mixed with B cells, another type of immune cell, in the pancreas tissue surrounding the islets. Given that B cells and T cells provide stimulatory signals to each other, these aggregates may promote inflammation and be a new therapeutic target.

Barlow et al. also wanted to understand why T cells target some islets more than others and why islet destruction is spatially organized. The team compared pancreatic areas with many inflamed islets to areas in the same donor where fewer islets were inflamed, finding that the cell composition differs. Interestingly, the types of cells that were different were not the same as those that were infiltrating islets. B cells, macrophages and T cells were the major cell types infiltrating islets, but the cells that varied outside islets were nerves, endothelial cells, and a third cell type that may have been innate lymphoid cells. These results indicate a crosstalk between the cells outside islets and those that infiltrate islets.

The results by Barlow et al. lay the groundwork for a better understanding of the biology underpinning how the immune system destroys β-cells in insulitis. The next steps would be to see if other cells in the islets can influence T cells and if diabetes could be delayed by inhibiting interactions between T cells and the relevant cells outside the islets. Moreover, it would be important to establish whether preserving IDO in endothelial cells could delay diabetes symptoms.

## Introduction

In type 1 diabetes (T1D), insulin-producing β-cells are killed by islet-infiltrating immune cells in a process called 'insulitis'. T1D results in a critical requirement for exogenous insulin and affects over eight million individuals world-wide with an estimated 0.5 million new diagnoses each year (*Gregory et al., 2022*).

Recently, the first immunotherapy for delaying T1D onset, teplizumab (a human anti-CD3 monoclonal antibody) was approved by the US Food and Drug Administration (*Hirsch, 2023*). However, this treatment and other immunotherapies help only a small fraction of patients and are significantly less effective after patients progress to overt T1D (*Herold et al., 2013*; *Perdigoto et al., 2019*; *Herold et al., 2019*; *Pescovitz et al., 2009*; *Orban et al., 2011*; *Orban et al., 2014*; *Bluestone et al., 2021*). A better understanding of T1D pathogenesis is essential to building on this progress.

One of the challenges of studying human T1D pathology is the availability of suitable tissue samples. Obtaining pancreatic biopsies raises the risk of surgical complications and the progressive nature of T1D would necessitate serial, longitudinal studies over time, which is prohibitive (*Krogvold et al., 2014*). Fortunately, the Juvenile Diabetes Research Foundation (JDRF) Network for Pancreatic

Organ Donors with Diabetes (nPOD) provides human pancreatic tissues from cadaveric donors for this study and many others (*Campbell-Thompson et al., 2012*; *Pugliese et al., 2014*). nPOD has enabled substantial progress towards characterizing the pathology of human T1D (*Wilcox et al., 2016*; *Arif et al., 2014*; *Leete et al., 2016*; *Martino et al., 2015*; *Korpos et al., 2021*).

Our understanding of key features of human T1D pathology remains limited. Although the cellular composition of insulitis, inflammation specifically of the islets, has been studied extensively, this has been done in separate studies looking at different tissue sections, prohibiting an understanding of how the numerous cellular programs in insulitis are coordinated throughout disease. This was recently addressed using Imaging Mass Cytometry (IMC), which uncovered alterations in β-cell phenotypes, immune composition, vascular density, and basement membrane that accompany T1D (*Damond et al., 2019*; *Wang et al., 2019*). However, these studies did not deeply phenotype islet-infiltrating CD8[+]T cells, believed to be major driver of β-cell elimination.

Recently, intriguing differences in the extra-islet spaces of T1D and healthy controls have been reported. First, the abundance of multiple types of immune cells outside islets are increased in T1D patients compared to non-T1D controls (*Rodriguez-Calvo et al., 2014*; *Campbell-Thompson et al., 2015*; *Bender et al., 2020*). Second, HLA-DR expression is increased on ductal cells in T1D tissue donors, hinting at a functional link with CD4[+]T cells (*Fasolino et al., 2022*). Third, peri-insulitis, the accumulation of immune cells outside islets, is observed in tissues from patients with T1D *Korpos et al., 2013*, indicating that not all T cells enter the pancreas directly via islet microvasculature (*Savinov et al., 2003*). Fourth, in human T1D, but less so in most animal models, islets in different regions of the pancreas are infiltrated at strikingly different rates for reasons that are unknown (*In't Veld, 2014*). This suggests that the extra-islet compartment could be responsible by governing the targeting of islets. Finally, tertiary lymphoid structures (TLSs) — dense aggregates of lymphoid cells indicative of local immune activation — are observed outside islets in T1D patients (*Korpos et al., 2021*).

In summary, analyzing both compartments simultaneously could help identify how these extra-islet factors influence islet pathogenesis. However, to date, multiplexed imaging studies have only examined islets. A comprehensive, spatially resolved cellular analysis of both compartments in T1D is lacking.

Here, we investigated the islet and extra-islet pancreas together. We used CO-Detection by indexing (CODEX) with an antibody panel targeting 54 antigens to samples from a cohort of T1D patients with insulitis as well as non-T1D individuals with and without islet-specific autoantibodies (AA- and AA+ respectively) obtained through the JDRF nPOD program. We analyzed approximately 2000 islets and broad swaths of the extra-islet tissue to evaluate local and distal spatial architecture. We then used pseudotime analysis to characterize insulitis sub-states based on the activation states of islet-infiltrating CD8[+]T cells. We further investigated the cellular changes in niches and lobules beyond islets. Our results implicate both the islet microenvironment and inflammation at distal sites within the pancreas in the progression of insulitis.

## Results

### Cohort curation, image acquisition, and cell annotation

The JDRF nPOD is a national registry of cadaveric pancreases donated by T1D patients that has transformed the ability of researchers to investigate the pathways underlying the progression of human T1D (*Campbell-Thompson et al., 2012*; *Pugliese et al., 2014*). Insulitis is present in most newly diagnosed T1D cases but in only a small fraction of total T1D cases, including those available from nPOD (*In't Veld, 2011*; *Atkinson and Mirmira, 2023*). Although nPOD had close to 200 T1D cases, at the time of our study, only 17 had documented insulitis. Of these, triple-immunohistochemistry for Insulin, Glucagon, and CD3 was performed. T1D and AA+ cases that had CD3 staining in islet or peri-islet spaces and tissue still available were selected for our study. The final cohort included two AA+ cases, eight T1D cases, and three non-T1D controls. Given that insulitis is not detected in non-T1D cases (*Bruggeman et al., 2021*), the blocks from controls were selected randomly. The T1D cases varied in the time between diagnosis and death from 0 years (diagnosed at death) to 6 years (*Figure 1A*, left). The causes of death were mostly unrelated to T1D complications (*Table 1*).

Large regions averaging 55 mm$^2$ were imaged with CODEX as previously described (*Schürch et al., 2020*; *Phillips et al., 2021a*; *Hickey et al., 2021*). Regions were selected to capture islets and the

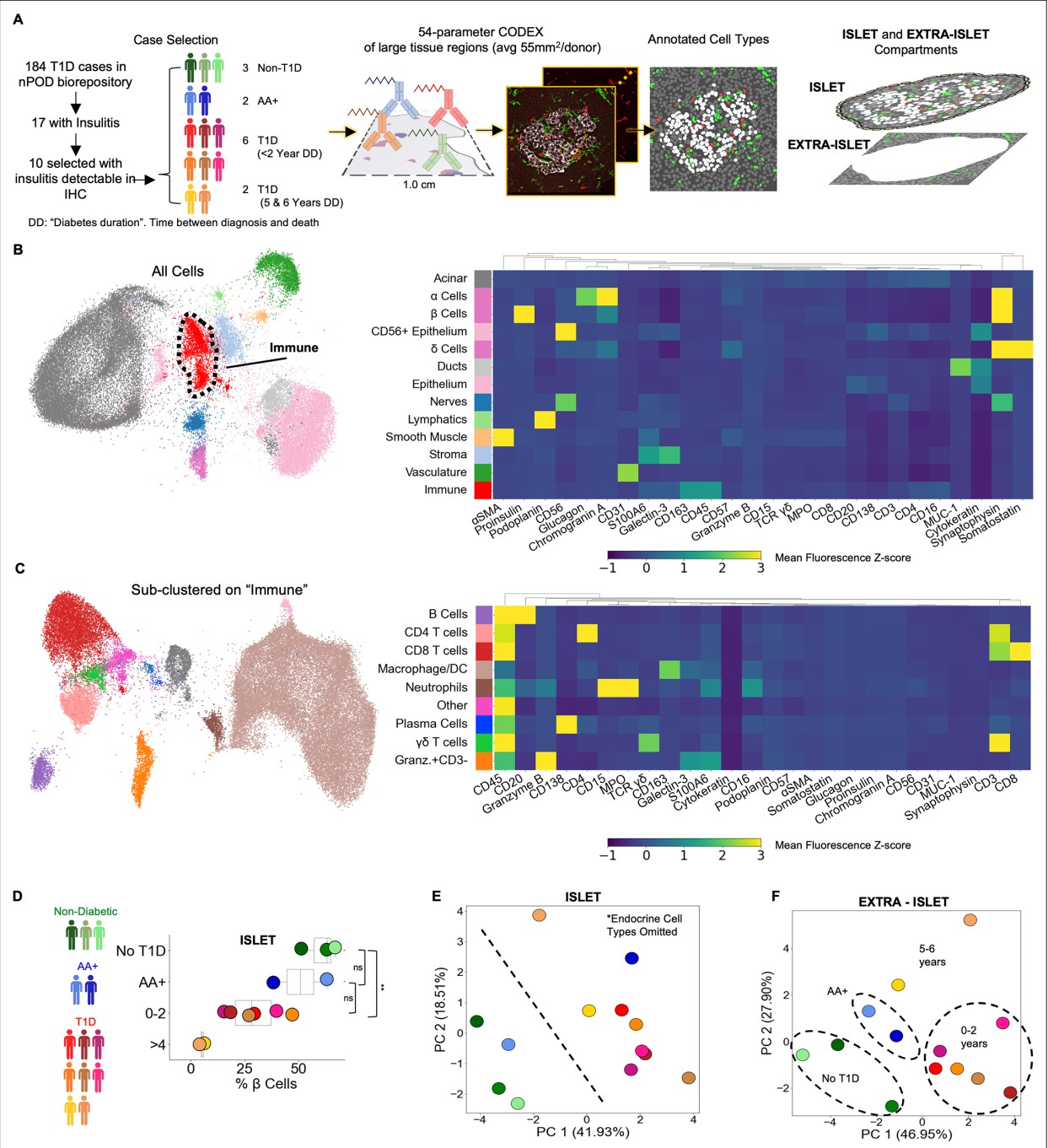

**Figure 1.** Profiling T1D pancreata with CODEX high-parameter imaging reveals alterations in the cellular composition of islet and extra-islet compartments. Left: Schematic of the workflow for selection of nPOD cases. Blues, greens, and reds indicate non-T1D, AA+, or T1D status, respectively. Center: Schematic for acquisition and processing of CODEX highly multiplexed imaging dataset. Right: Schematic of islet and extra-islet pancreatic regions. (**B**) UMAP and Leiden clustering of major cell types. Colors match those in the heatmap shown to the right. Heatmap of mean z-normalized marker expression in each cell type cluster. Only a subset of the markers used for the UMAP are included in the heatmap to facilitate visualization. A full description of the markers used for the clustering stages is available in *Table 2*. (**C**) UMAP of the immune population identified in (**B**) further clustered using additional immune markers as described in *Table 2*. Colors match those in heatmap shown in the heatmap to the right. The heatmap is generated in an identical manner as the heatmap in (**B**). (**D**) Frequency of β-Cells per donor determined by dividing the number of β-cells by the total number of β-cells, α-cells, and δ-Cells. Blues, greens, and reds indicate non-T1D, AA+, or T1D status, respectively. Significance was determined using the t-test (* p<0.05, ** p<0.01, *** p<0.001). (**D**) Principal component analysis of the islet compartment. The number of cells of each cell type (omitting α-, β-, and δ-cells) were divided by the number of endocrine cells to adjust for different islet areas. Blues, greens, and reds indicate non-T1D, AA+, or T1D status,

*Figure 1 continued on next page*

*Figure 1 continued*

respectively. (**E**) Principal component analysis of the extra-islet compartment. The number of cells of each cell type (omitting α-, β-, and δ-cells) were divided by the number of acinar cells to adjust for different areas imaged. Blues, greens, and reds indicate non-T1D, AA+, or T1D status, respectively.

The online version of this article includes the following figure supplement(s) for figure 1:

**Figure supplement 1.** Validation of cell annotations.

**Figure supplement 2.** The total number of endocrine cells measured in each donor.

**Figure supplement 3.** Changes in cellular abundance in Islet (top) and extra-islet (bottom) regions.

surrounding region simultaneously (*Figure 1A*, center and right). CellSeg was used to segment cell nuclei and quantify marker expression from CODEX images as previously described *Lee et al., 2022*. In total, our dataset consisted of $7.0 \times 10^6$ cells across all donors (ranging from $3.0 \times 10^5$ to $9.8 \times 10^5$ cells per donor). Twenty-one cell types were identified with Leiden clustering and manual merging and visualized using Uniform Manifold Approximation and Projection (UMAP; *Figure 1B*; *Table 2*). Endocrine cells were manually gated from UMAP embeddings derived from Proinsulin, Glucagon, and Somatostatin to identify β-cells, α-cells, and δ-cells, respectively. Immune cells were sub-clustered with the Leiden algorithm using immune-specific markers (*Figure 1C*; *Table 2*). To verify the accuracy of our annotations, we overlaid cell labels onto the original images (*Figure 1—figure supplement 1*). Of note, we could not accurately identify macrophage subsets or distinguish dendritic cells from macrophages due to the panel design, complex combinations of co-expression, and the difficulty in segmenting and quantifying markers on myeloid populations due to their morphology. Therefore, we refer to this cluster as 'macrophage/DCs' In addition, we identified a cell population that could not be definitively annotated that expressed high levels of CD45, CD69, Granzyme-B, and CD44, intermediate levels of CD16, S100A6, Galectin-3, and Hyaluronan, but not expressing CD3, CD20, CD56, CD57, CD15, or MPO. We confirmed from the raw images that CD3, CD4, and CD8 were not internalized, indicating activation, nor did these cells express other T cell activation markers CD45RA, CD45RO, PD-1, or LAG-3 (*Figure 1C*, *Figure 1—figure supplement 1* bottom right). This population could represent a type of innate lymphoid cell (*Dalmas et al., 2017*) and was labeled 'Granzyme-B+/ CD3-'.

**Table 1.** nPOD case information.

| Case ID | Donor Type | Age (years) | Diabetes Duration (years) | Cause of Death | Gender | Ethnicity | BMI | nPOD RRID |
|---------|-----------|-------------|---------------------------|----------------|--------|-----------|-----|-----------|
| 6267 | Autoab positive | 23 | NA | Anoxia | Female | Caucasian | 16.59 | SAMN15879321 |
| 6314 | Autoab positive | 21 | NA | Head Trauma | Male | Caucasian | 23.8 | SAMN15879368 |
| 6520 | T1D | 21.61 | 0 | Cerebrovascular/ | Male | Caucasian | 29.3 | SAMN18053203 |
| 6362 | T1D | 24.9 | 0 | Head Trauma | Male | Caucasian | 28.5 | SAMN15879415 |
| 6228 | T1D | 13 | 0 | Anoxia | Male | Caucasian | 17.4 | SAMN15879284 |
| 6209 | T1D | 5 | 0.25 | Cerebral edema secondary to DKA | Female | Caucasian | 15.9 | SAMN15879265 |
| 6371 | T1D | 12.5 | 2 | Cerebral edema | Female | Caucasian | 16.6 | SAMN15879424 |
| 6480 | T1D | 17.18 | 2 | DKA | Male | Caucasian | 27.1 | SAMN15879533 |
| 6195 | T1D | 19.3 | 5 | Head Trauma | Male | Caucasian | 23.7 | SAMN15879251 |
| 6323 | T1D | 22 | 6 | Anoxia | Female | Caucasian | 24.7 | SAMN15879377 |
| 6389 | No diabetes | 18.6 | NA | Head Trauma | Male | Caucasian | 20.9 | SAMN15879442 |
| 6179 | No diabetes | 20 | NA | Head Trauma | Female | Caucasian | 20.7 | SAMN15879235 |
| 6386 | No diabetes | 14 | NA | Head Trauma | Male | Caucasian | 23.9 | SAMN15879439 |

**Table 2.** Markers used for cell type identification.

Channels in the 'Both' column were used for clustering all cells and specifying immune cells. Channels in the 'All' column were only used for clustering all cells and UMAP in *Figure 1B*. Channels in 'Immune' columns were only used for sub-clustering immune cells and UMAP in *Figure 1C*. Channels in 'Endocrine' column were used for sub-clustering endocrine populations. Channels in 'Unused' column were not included in the clustering or UMAP step because they were either too weak to aid clustering or were expressed on multiple cell-populations and confounded cell type identification.

| Both | 'All' UMAP | 'Immune' UMAP | Endocrine Cells | Unused [§] |
|---|---|---|---|---|
| Channel 2 Blank* | alphaSMA | VISTA | Glucagon | CD44 |
| Channel 3 Blank* | Synaptophysin | TCR g/d | Insulin | CD45RA |
| Channel 4 Blank* | Podoplanin | MPO | Proinsulin | CD45RO |
| S100A6 | PD-L1 [‡] | HLA-DR | Somatostatin | CollV |
| Hoechst[†] | NaKATPase | FOXP3 | | HABP |
| Granzyme B | MUC-1 | CD8 | | HLA-ABC |
| Galectin-3 | Draq 5 | CD69 | | ICOS |
| CD68 | Cytokeratin | CD4 | | IDO |
| CD57 | Chromogranin A | CD206 | | Ki67 |
| CD56 | CD31 | CD16 | | Lag3 |
| CD45 | | CD11c | | OX40 |
| CD3 | | BCL-2 | | PD-1 |
| CD20 | | | | TOX |
| CD163 | | | | |
| CD15 | | | | |
| CD138 | | | | |

*Channel 2–4 Blanks used for identifying autofluorescent cells. A cycle was run without adding fluorescent oligonucleotides.
[†]Hoechst and Draq 5 were both used as Draq 5 gives slightly more uniform staining which improves segmentation.
[‡]PD-L1 did not detect any positive myeloid cells or β-cells but stained nerve cells very brightly. Therefore, it was still useful to include.
[§]'Unused' were used in other places in the manuscript but not for cell type annotation.

## Islet- and extra-islet regions are altered in T1D

We first sought to identify cellular changes in T1D within islets specifically. Previous reports observed that insulin-containing islets are significantly more common in recent-onset T1D cases than cases with diabetes durations of greater than one year (*In't Veld, 2011*; *Campbell-Thompson et al., 2016*; *Richardson and Pugliese, 2022*). Similarly, we found that samples from patients who had been diagnosed with T1D for 0–2 years had significantly reduced β-cell frequencies compared to non-T1D controls. Furthermore, samples from subjects with disease durations of 5–6 years had minimal remaining β-cell mass (*Figure 1D*). Whereas one AA+ case had β-cell mass comparable to those of cases with disease duration of 0–2 years, the other AA+ case was comparable to non-T1D controls (*Figure 1D*). The total islet area imaged was comparable across all donors (*Figure 1—figure supplement 2*).

Next, we investigated how the abundances of non-endocrine cell types inside islets differed across donors. We performed Principal Component Analysis (PCA) on the donors using the frequencies of non-endocrine cell types located in islets. Donors were clearly separated into two groups by the first two principal components; one group included all T1D cases and one AA+ case and the second included all non-T1D cases and the other AA+ case (*Figure 1E*). In this analysis, we did not consider β-cells, α-cells, and δ-cells. Thus, donors were stratified by disease duration strictly according to the abundances of immune and other pancreatic, non-endocrine cell types in the islets.

We next considered only cells located outside islets. Again, donors were clearly separable by the first two principal components (*Figure 1F*). The first principal component separated cases with times since diagnosis between 0 and 2 years from non-T1D, AA+, and cases with diabetes durations of 5–6 years (*Figure 1F*). The second principal component separated cases with diabetes durations of 5–6 years from the rest (*Figure 1F*). Therefore, both the islet and extra-islet spaces of T1D and non-T1D cases were distinct.

Many cell types were increased in T1D cases with times since diagnosis of 0–2 years relative to non-T1D controls (*Figure 1—figure supplement 3*). In T1D cases with times since diagnosis of 5–6 years, the abundance of different cell types either remained higher than non-T1D controls or returned to baseline (*Figure 1—figure supplement 3*). This trend was present in both islet and extra-islet regions. These data demonstrate that the immune activity between the islet and extra-islet compartments are coordinated.

## Pseudotemporal reconstruction of islet pathogenesis identifies a conserved trajectory of insulitis

In human T1D, β-cell destruction does not occur simultaneously across all islets and even neighboring islets can be at different stages of destruction (*Damond et al., 2019*; *In't Veld, 2014*; *In't Veld, 2011*). We therefore used pseudotime analysis to infer the most likely progression of a single islet through disease space (*Damond et al., 2019*). To develop a pseudotemporal map, we quantified the cellular composition of each islet, including cells within 20 µm of the islet's boundary, and applied the pseudotime algorithm PArtition-based Graph Abstraction (PAGA) (*Wolf et al., 2019*; *Figure 2A*; *Figure 2B*). PAGA was selected because it is a high-performing algorithm able to identify multiple trajectories, if they exist, while making minimal assumptions about the true structure (*Saelens et al., 2019*).

As expected, islets from different donor groups (no T1D, AA+, T1D) had different distributions across pseudotime (*Figure 2B*; *Figure 2C*). In the PAGA map, a continuum is apparent from islets abundant in insulin-expressing β-cells on the left of the map to islets depleted in β-cells on the right (*Figure 2D*, *Figure 2E*, *Figure 2F*, top row). PAGA uses Leiden clustering internally, enabling the following regions of the pseudotime map to be labeled objectively: (1) Islets with low pseudotime values on the left of the map (PAGA-internal Leiden clusters 0 and 5 in *Figure 2—figure supplement 1*) were labeled 'Normal' even if they originated from T1D donors. (2) Islets in the middle of the map (PAGA-internal Leiden clusters 6, 2, and 8 in *Figure 2—figure supplement 1*) were elevated in HLA-ABC (MHC Class I) expression, CD8+T cells, and macrophage/DCs (*Figure 2D*, *Figure 2E*, *Figure 2F*, rows 2–4) and were labeled 'Inflamed'. (3) Islets with late pseudotime values on the right of the map (PAGA-internal Leiden clusters 1, 3, 7, and 4 in *Figure 2—figure supplement 1*) were devoid of β-cells and were labeled 'Insulin-Depleted' (*Figure 2D*, *Figure 2E*, *Figure 2F*, top row).

In addition, islets lacking β-cells occasionally contained CD8+T cells and were labeled 'Insulin-Depleted + Immune Islets' (*Figure 2D*, *Figure 2E*, *Figure 2F*, rows 2–4). The presence of these islets suggests that signals retaining CD8+T cells in islets linger after β-cells die. The distribution of all cell types across pseudotime is reported in *Figure 2—figure supplement 2*.

Islets from non-T1D controls and one of the AA+ donors (6314) were primarily in the Normal group to the left of the map (*Figure 2C*; *Figure 2—figure supplement 3*). Islets from subjects who had T1D for of 5–6 years (cases 6195 and 6323) were primarily in the Insulin-Depleted group to the right of the map (*Figure 2C*, *Figure 2—figure supplement 3*). Islets from the remaining T1D donors and the other AA+ donor were distributed broadly throughout the map (*Figure 2C*, *Figure 2—figure supplement 3*).

We quantified the fraction of each cell type in swaths at varying distances from the islet edge. We found that for B cells, CD4+T cells, CD8+T cells, macrophage/DCs, neutrophils, and plasma cells, the fraction of the given cell type in the islet relative to outside the islets increased (*Figure 2—figure supplement 4*), demonstrating that the inflammation in islets was distinct from the inflammation in the extra-islet tissue. Thus, immune cells were targeting islets specifically.

Together, these results illustrate a single, non-branching progression from Normal Islets to Insulin-Depleted Islets via Inflamed Islets, consistent with previous pseudotime analyses (*Damond et al., 2019*).

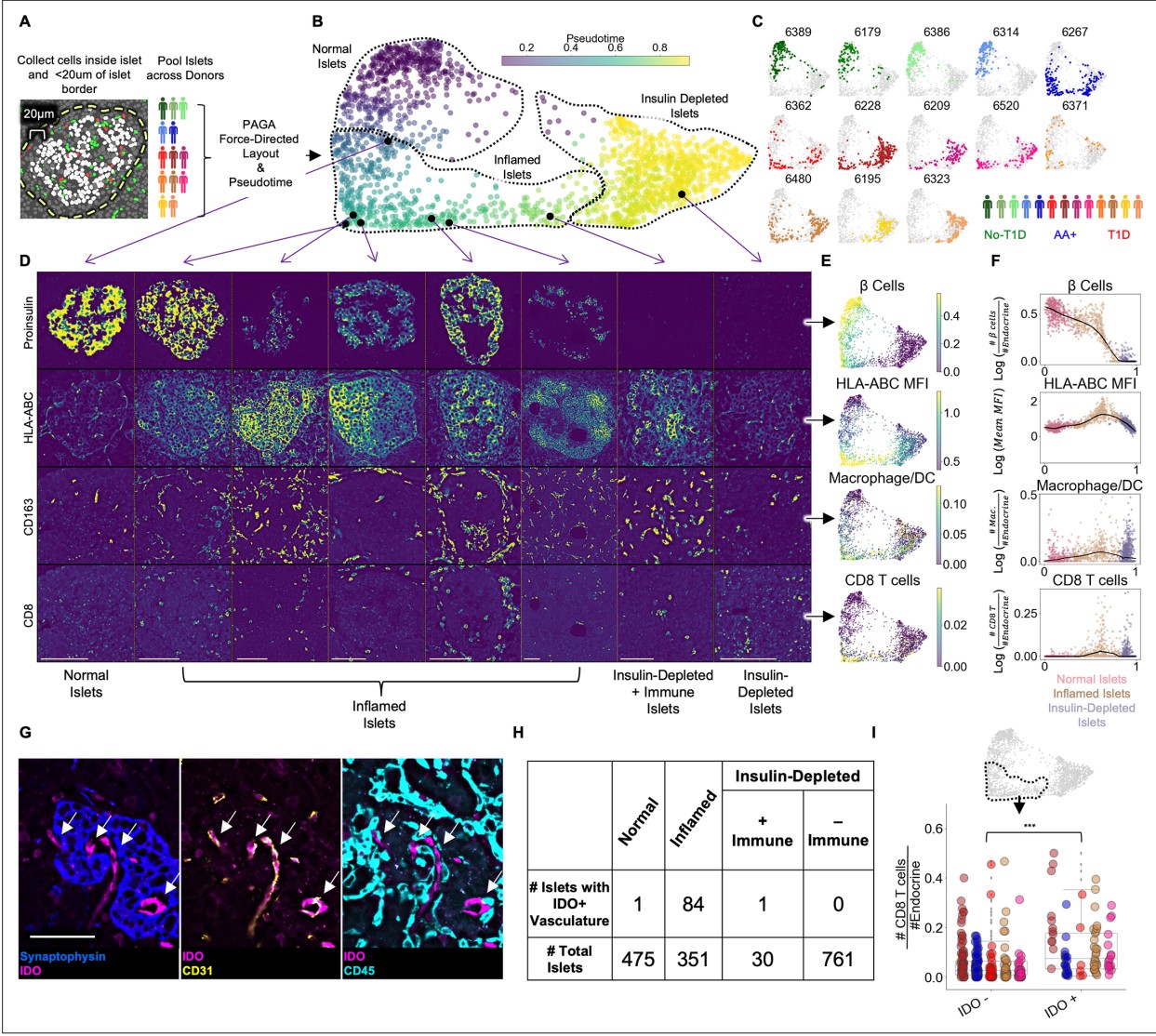

**Figure 2.** Pseudotemporal reconstruction of insulitis identifies IDO on islet vasculature. (**A**) Schematic of islet segmentation and quantification of islet cellular composition. (**B**) PAGA-force directed layout of islets colored by pseudotime. Each point represents an islet. Each islet's color reflects the pseudotemporal distance from the centroid of non-T1D islets. Representative islets from different stages of pseudotime are indicated with black points and their raw images are depicted in (**D**). Normal, Inflamed, and Insulin-Depleted groups were obtained by merging the clusters output by the PAGA algorithm (***Figure 2—figure supplement 1***). (**C**) Islet distribution across pseudotime for each donor. The titles indicate nPOD case IDs as in ***Table 1***. The frequency of islets from each donor in each stage of islet pseudotime is reported in ***Figure 2—figure supplement 3***. (**D**) Images of Proinsulin, HLA-ABC, CD163, and CD8 staining in islets representative of different points along pseudotime as indicated in B. Scale bars (lower left of each column) indicate 100 µm. (**E**) Quantification of selected features across pseudotime overlaid onto the PAGA force-directed layout. For β-cells, macrophage/DCs, and CD8+T cells, the values correspond to log(# cells/# endocrine cells). For HLA-ABC, the mean HLA-ABC expression for each cell in the islet was computed and log transformed. (**F**) Quantification of selected features across pseudotime. For β-cells, macrophage/DCs, and CD8+T cells, the values correspond to log(# cells/# endocrine cells). For HLA-ABC, the mean HLA-ABC expression for each cell in the islet was computed and log transformed. Color legend: Normal islets: pink; Inflamed islets: brown; Insulin-Depleted islets: purple. Black points demarcate LOWESS regression. (**G**) Representative image of an islet from the Inflamed group stained with IDO and, from left to right, Synaptophysin, CD31, and CD45. Arrows indicate IDO+/CD31+ vasculature. Right shows that IDO+ cells are negative for CD45 and therefore, are not immune cells associated with vasculature. Scale bar (bottom left image) indicates 50 µm. (**H**) Distribution of IDO expression on islet vasculature across pseudotime. (**I**) Association of IDO+ islet vasculature with islet CD8+T cell density. The y-axis corresponds to the number of CD8+T cells per endocrine cell per islet. CD8+T cell counts were normalized to adjust for islet size. The x-axis indicates whether islets contain IDO+ vasculature. Each color corresponds to a donor. All donors with detectable IDO+ islet vasculature are displayed: 6480, 6267, 6520, 6228, and 6362. Colors are same as in (**H**). IDO+ vasculature was manually quantified. For combined donors, significance was determined with a mixed-effect linear model, p = 1.5 x 10–12 (Satterthwaites's method lmerTest R package). The complete breakdown per donor is reported in ***Figure 2—figure supplement 6***.

*Figure 2 continued on next page*

*Figure 2 continued*

The online version of this article includes the following figure supplement(s) for figure 2:

**Figure supplement 1.** Leiden clustering computed by PAGA algorithm internally.

**Figure supplement 2.** The density of each cell type per islet across pseudotime.

**Figure supplement 3.** The number of islets of each stage of pseudotime and the total number of cells per case.

**Figure supplement 4.** For each cell type, the frequency of that cell type inside islets, within 100 μm of the islet edge, and 100 μm–150 μm from the islet edge was quantified.

**Figure supplement 5.** Frequency of IDO on vasculature at different distances from islets.

**Figure supplement 6.** CD8⁺T cell, Macrophage, γ/δ T cell, and CD4⁺T cell abundance in IDO⁺ and IDO⁻ islets.

## IDO expression on islet vasculature is linked to T cell infiltration

We observed islets in which CD31⁺ vasculature stained positive for indoleamine 2, 3-dioxygenase 1 (IDO). In the tumor microenvironment, IDO is commonly expressed by myeloid cells and suppresses CD8⁺T cell activity through multiple mechanisms, including the induction of FOXP3⁺ regulatory T cells and the inhibition of CD8⁺T cell function (*Munn and Mellor, 2016*). In islets, IDO did not co-stain with CD45⁺ immune cells adjacent to vasculature (*Figure 2G*). We did not observe IDO expression in endocrine or other cell types in islets or in vasculature or any cell type outside islets (*Figure 2—figure supplement 5*). We manually quantified vascular expression of IDO in islets throughout pseudotime and found that all but two IDO⁺ islets were in the Inflamed group (*Figure 2H*). Therefore, IDO expression by islet vasculature was tightly associated with insulitis.

A potent inducer of IDO expression is interferon-γ (IFN-γ), a cytokine highly expressed by activated T cells and macrophages (*Munn and Mellor, 2016*). Therefore, we hypothesized that IDO expression was induced by infiltrating immune cells during insulitis. We compared the frequency of CD8⁺T cells and macrophage/DCs in islets from the Inflamed group with and without IDO⁺ vasculature and found that CD8⁺T cells were significantly more abundant in islets with IDO⁺ vasculature than islets without IDO⁺ vasculature (*Figure 2I*). In addition, the expression of HLA-ABC, another interferon-stimulated gene, was higher in β-cells in IDO⁺ islets than in β-cells in IDO⁻ islets (*Figure 2—figure supplement 6*). However, the abundance of γδ-T cells and CD4⁺T cells was less strongly associated with IDO⁺ vasculature and the abundance of macrophage/DCs was not significantly associated with IDO⁺ vasculature (*Figure 2—figure supplement 6*).

In summary, IDO expression by islet vasculature is positively associated with T cell infiltration and may be an immunoregulatory checkpoint in T1D.

## Insulitis has sub-states, defined by functional states of CD8⁺T cells

CD8⁺T cells are a major component of insulitis (row 4 of *Figure 2E* and *Figure 2F*) and are capable of directly and indirectly killing β-cells. A comprehensive description of the activation profiles of CD8⁺T cells could provide insight into their roles in T1D pathogenesis. To obtain extremely high-quality marker quantification, we trained a neural network on manually labeled images of single T cells (*Figure 3A*, *Figure 3—figure supplement 1*). Using our neural network, we quantified the expression of T cell markers on islet CD8⁺T cells (*Figure 3B*).

PD-1, TOX, CD45RO, CD69, and CD44, markers of antigen experience, were the markers most frequently expressed by islet-infiltrating CD8⁺T cells (*Figure 3B*, *Figure 3—figure supplement 2*). CD8⁺T cells expressing CD45RA (which are either naive or terminally differentiated effector memory cells [TEMRA]) were detectable in islets, as previously reported (*Damond et al., 2019*; *Figure 3B*). In addition, we observed a rare population of CD45RO⁺/CD8⁺T cells co-expressing LAG-3, Granzyme-B, and ICOS (P bottom clade). Lastly, a rare population of CD57⁺/CD8⁺T cells was present but these cells rarely co-expressed LAG-3, Granzyme-B, or ICOS (*Figure 3B* top clade). These populations resemble the two exhausted T cell populations identified in the peripheral blood of T1D patients that were associated with responsiveness to alefacept *Diggins et al., 2021*. Therefore, the activation profiles of islet-infiltrating T cells are heterogeneous.

We reasoned that the islet microenvironment may dictate the activation state of CD8⁺T cells by specifically recruiting T cells of a particular state or inducing changes after they enter the islet. If so, islets would contain specific combinations of CD8⁺T cell states. To interrogate this, we performed UMAP only on Inflamed islets, using the frequencies of CD8⁺T cells expressing each functional marker.

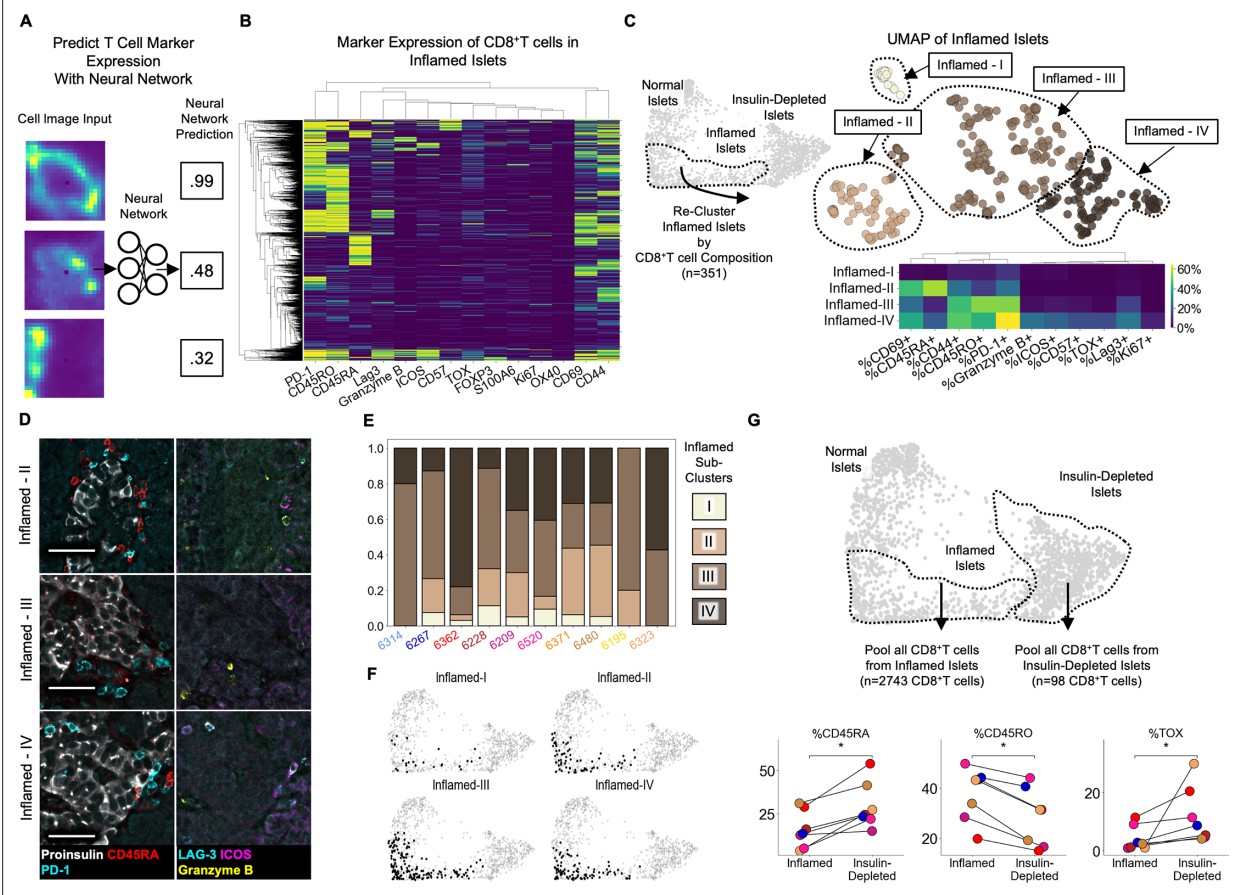

**Figure 3.** Insulitis has sub-states, characterized by CD8+T cell functionality (**A**) Schematic of marker-quantification with a ResNet50 neural network. Cell images are input, and the neural network outputs a number between 0 and 1 indicative of the level of confidence that the cell expresses that marker with 1 indicating the highest confidence. (**B**) Heatmap of all 2855 Inflamed Islet CD8+T cells, hierarchically clustered according to marker expression predicted by the neural network. (**C** Top) UMAP of Inflamed Islets based on frequencies of markers on CD8+T cells in islets. (**C** Bottom) Mean frequencies of each marker on CD8+T cells in islets of each inflamed sub-cluster. (**D**) Representative images of islets from each subcluster with associated immune markers. Scale bars indicate 50 μm. (**E**) Frequencies of islets from each subcluster per donor in AA+ and T1D samples. Color indicates subcluster as in panel C. (**F**) Distribution of the islets of Inflamed-I through -IV on the PAGA force-directed layout shown in **Figure 2B**. (**G**) Differences in marker expression frequencies between CD8+T cells in islets from the Inflamed group and from the Insulin-Depleted + Immune group. T cells from all islets of the specified groups were pooled within each donor to compute the frequencies of marker expression. Significance was determined using the Wilcoxon signed-rank test (* p<0.05, ** p<0.01, *** p<0.001) and was not corrected for multiple hypothesis testing.

The online version of this article includes the following figure supplement(s) for figure 3:

**Figure supplement 1.** Validation of neural network for detecting expression of T cell markers.

**Figure supplement 2.** Frequency of functional markers on CD8+T cells inside islets.

**Figure supplement 3.** Frequency of functional markers on CD8+T cells at different distances from islets.

**Figure supplement 4.** Association of islet features with Inflamed-I through -IV.

We identified four inflamed sub-clusters, I–IV, (**Figure 3C** top). Here, the term 'sub-cluster' is used to highlight that these groups were all contained within the previously defined 'Inflamed' cluster and the roman numerals do not imply a temporal ordering. Inflamed-I contained only CD8+T cells that did not express any of the functional markers analyzed (**Figure 3C** bottom, top row). Inflamed-II was characterized by a high frequency of CD45RA+/CD8+T cells (**Figure 3C** bottom, second row from top and **Figure 3D** top row). Inflamed-III was characterized by a low frequency of CD45RA+ cells and high frequency of CD45RO+ and PD-1+ cells (**Figure 3C** bottom, third row from top and **Figure 3D** middle row). Inflamed-IV was characterized by an enrichment of CD8+T cells expressing CD57, LAG-3, ICOS, Granzyme-B, PD-1 or CD45RO (**Figure 3C** bottom, bottom row and **Figure 3D** bottom row). In summary, the phenotypes of CD8+T cells are coordinated across islets.

## Regulation of insulitis sub-states by the islet microenvironment

To identify cellular or molecular factors that regulate the state of CD8+T cells in islets, we first inspected the distribution of inflamed sub-clusters in each patient. Each donor possessed islets that belonged to multiple inflamed islet sub-clusters (*Figure 3E*). Therefore donor-level factors such as genetics, the location within the pancreas (i.e. head, body, or tail), or time since T1D onset, are not associated. Instead, these insulitis sub-states are conserved among T1D patients.

Next, we asked if each T cell marker is enriched in CD8+T cells in islets compared to CD8+T cells in the peri-islet and exocrine space. We computed the frequencies of each CD8+T cell state inside islets of each inflamed sub-cluster and in separate swaths 0–25 µm, 25–50 µm and 50–100 µm away from the islets (*Figure 3—figure supplement 3*). We found that for islets of Inflamed-II, -III, and -IV, functional markers characterizing their CD8+T cells were expressed more frequently inside than in the surrounding tissue areas. Although the different functional markers were all enriched on islet-infiltrating T cells compared to T cells outside islets, the degree of this enrichment varied across the markers measured (*Figure 3—figure supplement 3*). The markers of Inflamed-IV, LAG-3, ICOS, and Granzyme-B, were highly enriched inside islets (~20% of islet-infiltrating CD8+T cells vs <5% of extra-islet CD8+T cells). CD45RA and CD69 in Inflamed-II were slightly less enriched in islets (~28% of islet-infiltrating and ~20% of extra-islet CD8+T cells). CD45RO and PD-1 in Inflamed-III islets were the most abundant but had the least enrichment in islets (~45% of islet-infiltrating CD8+T cells vs ~35% of extra-islet CD8+T cells; *Figure 3—figure supplement 3*). This demonstrated that the specific differences in the compositions of CD8+T cell states in different islets were attributable to the islet micro-environment and not the surrounding extra-islet spaces.

Although macrophage/DCs are abundant in islets from the Inflamed group (*Figure 2E*, *Figure 2F*) and can interact with T cells by presenting antigen or secreting cytokines, neither the expression of markers of macrophage/DC activity nor macrophage/DC abundance was significantly associated with any of the inflamed sub-clusters (*Figure 3—figure supplement 4*). Similarly, no other cell type nor the vascular expression of IDO was linked to CD8+T cell programs in islets (*Figure 3—figure supplement 4*). Accordingly, the four inflamed sub-clusters had identical distributions throughout the original PAGA force-directed layout (*Figure 3F*). Therefore, the activation states of islet-infiltrating T cells are independent of the abundance of any of the other cell types we could identify.

Lastly, we compared CD8+T cells in islets from the Insulin-Depleted + Immune group to CD8+T cells in islets from the Inflamed group. Insulin-Depleted + Immune islets contained a higher frequency of CD45RA+/CD8+T cells and a lower frequency of CD45RO+/CD8+T cells than Inflamed islets (*Figure 3G*). TOX was expressed by a higher frequency of CD8+T cells in Insulin-Depleted + Immune islets than CD8+T cells in Inflamed islets (*Figure 3G*). Importantly, CD45RA and TOX were never co-expressed on the same CD8+T cell (*Figure 3B*). These data indicate that the activation state or persistence of these two populations in islets depends on insulin expression.

## Vasculature, nerves, and Granzyme-B+/CD3- cells outside islets are associated with lobular patterning

In the pancreas, islets are grouped within lobules. In some donors, the destruction of islets in T1D exhibits a strong lobular patterning: that is, insulin-positive, immune-infiltrated, and insulin-negative islets are each primarily found in different lobules (*Gepts, 1965*). However, other donors are on the other end of this spectrum and have lobules comprised of a random mixture of islets from different stages of pathogenesis.

It is unknown why some cases exhibit lobular patterning. One explanation could be an islet-intrinsic mechanism where the expression of programs sensitizing β-cells to immune-killing (i.e. stress) are correlated across islets in the same lobule. Alternatively, it could be mediated by cells outside islets if they facilitate extravasation into the lobule or trafficking from one afflicted islet to the next.

To systematically investigate lobular patterning in T1D, we used a neural network to segment lobules and assign each single cell and islet to its lobule. We first quantified the degree of lobular patterning within each donor using the intra-class correlation coefficient (ICC). The ICC ranges from 0 to 1 where cases with values closer to 1 have islets whose states are more synchronized within lobules (*Figure 4A*). Islets of non-T1D cases and 6314, 6195, and 6323 did not have appreciable variability in their pseudotimes, but in the remaining cases, the ICCs ranged from 0.17 to 0.74 (*Figure 4B*). This highlights that the magnitude of lobular patterning ranges widely across T1D cases with insulitis.

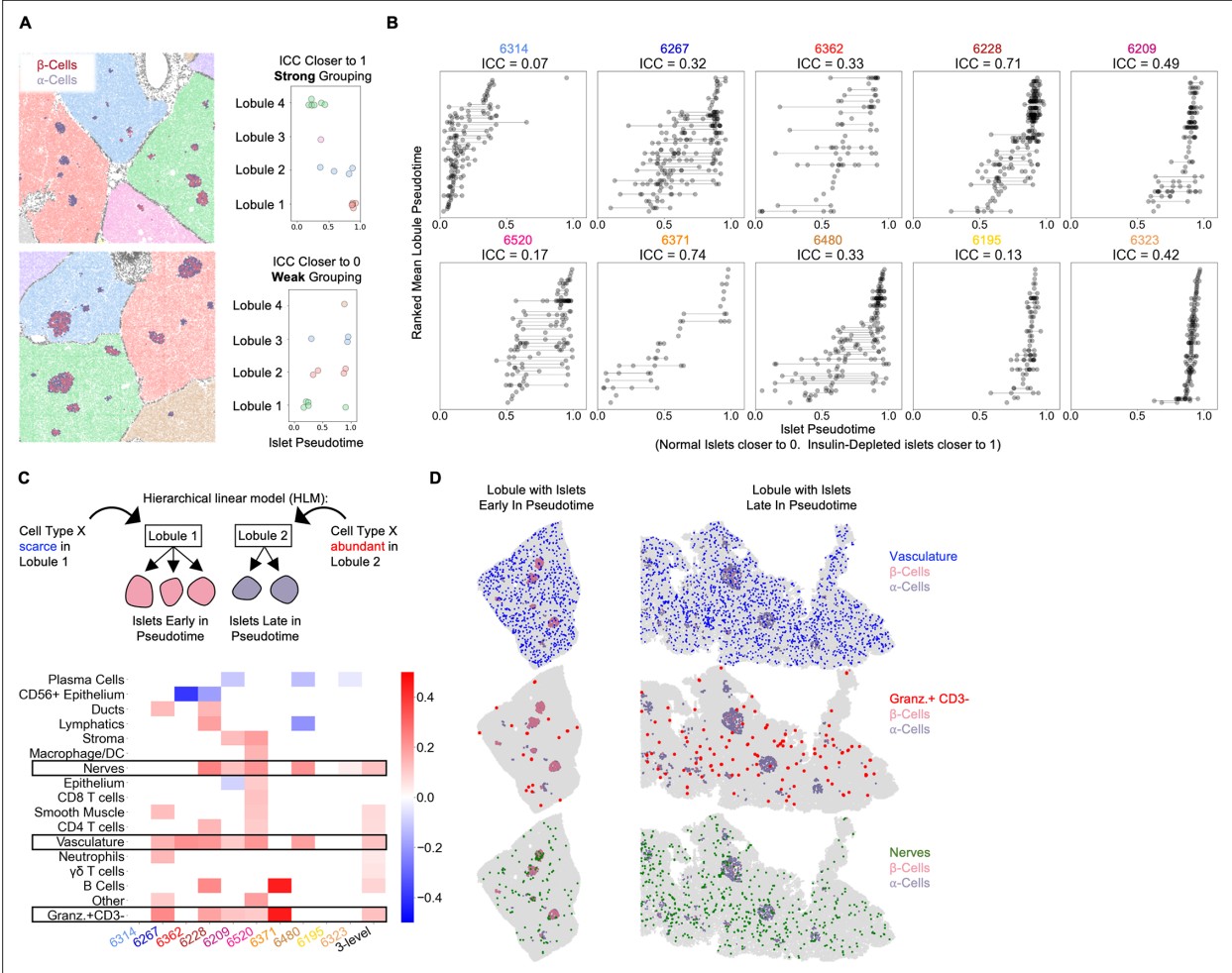

**Figure 4.** Vasculature, nerves, and Granzyme-B⁺/CD3⁻ cells in the extra-islet pancreas are associated with the lobular patterning of islet pathogenesis (**A**) A schematic of the method for quantifying lobular patterning of insulitis. Lobules were segmented and colored accordingly. The islets are colored according to their composition of β-cells and α-cells. Top: A region from case 6228 with a strong lobular grouping effect and an ICC closer to 1. Bottom: A region from case 6267 with a weak lobular grouping effect and an ICC closer to 0. (**B**) Lobular patterning of insulitis within each donor. Each point represents an islet. The x-axis represents the islet pseudotime. The y-axis is ordinal, representing the rank of each lobule according to the mean pseudotime of its islets. Violin plots per lobule are overlaid. ICC: Intraclass correlation coefficient. (**C**) Cell types associated with lobular patterning. Top: Schematic of the hierarchical linear model. Cells in islets were omitted when computing the lobular abundance of each cell type. Bottom: Coefficients of two-level models trained on each donor separately (columns labeled by donor) and a three-level model (right column). Color corresponds to the coefficient and features with p>0.05 are white. Significance was determined using Satterthwaites's method in the lmerTest R package. No adjustment for multiple hypothesis testing was applied. (**D**) Visualization of vasculature (top), Granzyme-B/CD3⁻ cells (middle), and nerves (bottom) in two lobules. The left lobule represents lobules with islets earlier in pseudotime and a lower abundance of the given cell type in the lobule. The right lobule represents lobules with islets late in pseudotime and a greater abundance of the given cell type in the lobule.

The online version of this article includes the following figure supplement(s) for figure 4:

**Figure supplement 1.** Changes in cell types identified by HLM in insulitis.

We employed hierarchical linear modeling (HLM), a statistical framework designed to identify relationships between levels of multi-level data. HLMs are standard in fields where multi-level data are common such as Education, in which students are grouped into classrooms, which are grouped into schools (*Gelman, 2021*) and have been applied in biomedical settings (*Jerby-Arnon and Regev, 2022*; *Yi et al., 2019*). We were interested in cell types if their abundance in a lobule correlated with the lobule's average islet pseudotime. Importantly, we omitted cells within islets from the calculation of a cell type's lobular abundance because we were interested in identifying features in the extra-islet space that were associated with islet destruction. For each cell type, we estimated the effect of its total abundance in a lobule (the number of cells divided by the number of acinar cells to normalize for

lobule area) on the pseudotimes of islets in that lobule. We performed this analysis in two-level HLMs for each donor and a three-level HLM considering all donors together (*Figure 4C*).

We identified three cell types — vasculature, nerves, and Granzyme-B$^+$/CD3$^-$ cells — that were significantly associated with lobules across multiple T1D tissue donors. All three were more abundant in lobules with islets late in pseudotime (*Figure 4C* boxed rows, *Figure 4D*). Samples from cases 6323 and 6195 which had very few insulin-containing islets had increased abundances of vasculature, Granzyme-B$^+$/CD3$^-$ cells, and nerves in their extra-islet spaces compared to non-T1D controls (*Figure 1—figure supplement 3*), indicating these changes persist through the point when the entire tissue is afflicted. In addition, vasculature, Granzyme-B$^+$/CD3$^-$ cells, and nerves were increased in Inflamed islets compared to Normal islets indicating that they may serve a role in islets in addition to their role in the extra-islet compartment (*Figure 4—figure supplement 1*). Note that *Figure 4—figure supplement 1* differs from *Figure 1—figure supplement 3* because islets are broken up according to pseudotime, not donor, and not all islets in T1D donors are undergoing insulitis.

Counterintuitively, although CD8$^+$T cells and macrophage/DCs were higher in the extra-islet compartments of T1D cases vs non-T1D cases (*Figure 1—figure supplement 3*), they were not associated with lobular patterning (they were not more abundant in the extra-islet space of lobules with more advanced insulitis; *Figure 4C*). These data raise the possibility that vasculature, Granzyme-B/CD3- cells, and nerves outside islets help predispose lobules to insulitis or are affected by extensive insulitis.

## Immature tertiary lymphoid structures are enriched in subjects with T1D

We hypothesized that T cells' interactions with certain cell types in specific areas of the pancreas may be important for their functionality. Therefore, we identified cellular neighborhoods (CNs; *Schürch et al., 2020*; *Bhate et al., 2022*), tissue regions that are homogeneous and have defined cell type compositions. To identify CNs, single cells were clustered according to the cell type composition of their twenty nearest spatial neighbors and automatically annotated with the names of enriched cell types (*Figure 5A*, See Materials and methods). This resulted in 75 CNs. Throughout the manuscript, CNs are referred to with the nomenclature (Cell Type A|Cell Type B|…) to indicate all the cell types that are enriched in them (See Materials and methods).

Next, we identified CNs that were more abundant in T1D than non-T1D tissues (*Figure 5B*). The top three CNs (fold change of abundance in T1D relative to abundance in non-T1D) were (CD8$^+$T cells|B cells), (Macrophage|Stromal Cells|B cells), and (Vasculature|B cells) (*Figure 5B*, *Figure 5C*). We asked whether these three CNs were commonly adjacent to each other as this could indicate that they act as components of a larger structure (*Bhate et al., 2022*). Measuring the frequency with which the three B cell CNs were adjacent to each other throughout the tissues demonstrated that the (CD8$^+$T cells|B cells) CN is predominantly found adjacent to both the other CNs but that (Macrophage|Stroma|B cells) and (Vasculature|B cells) are less commonly adjacent to each other (*Figure 5D*).

We next asked whether these CN assemblies corresponded to either peri-vascular cuffs (*Agrawal et al., 2013*; *Wekerle, 2017*) or tertiary lymphoid structures (TLSs) (*Korpos et al., 2021*; *Agrawal et al., 2013*; *Rovituso et al., 2016*), as these are two B cell-rich structures commonly present in autoimmune conditions. Although the (CD8$^+$T cells|B cells) CN was adjacent to vessels (*Figure 5D*, *Figure 5E*), it was not in the perivascular space, as is the case with perivascular cuffs (*Figure 5E*). In our samples, the (CD8$^+$T cells|B cells) CN did not have segregated T cell and B cell zones as seen in mature TLSs, consistent with previous reports (*Korpos et al., 2021*).

In summary, the (CD8$^+$T cells|B cells) CN is more abundant in T1D tissues from patients with diabetes durations of 0–2 years compared to non-T1D tissues and T1D tissue from patients who had T1D for more than 4 years.

## Immature tertiary lymphoid structures are in the extra-islet pancreas, and are enriched in CD45RA$^+$ /CD8$^+$ T cells

We next asked whether the (CD8$^+$T cells|B cells) CN had high endothelial venules (HEV), specialized blood vessels that are commonly found in TLSs that enable naive lymphocytes to extravasate into peripheral tissues. We observed expression of peripheral lymph node addressin (PNAd), an HEV marker, in the vessels associated with the (CD8$^+$T cells|B cells) CN (*Figure 5E* left image) but not in

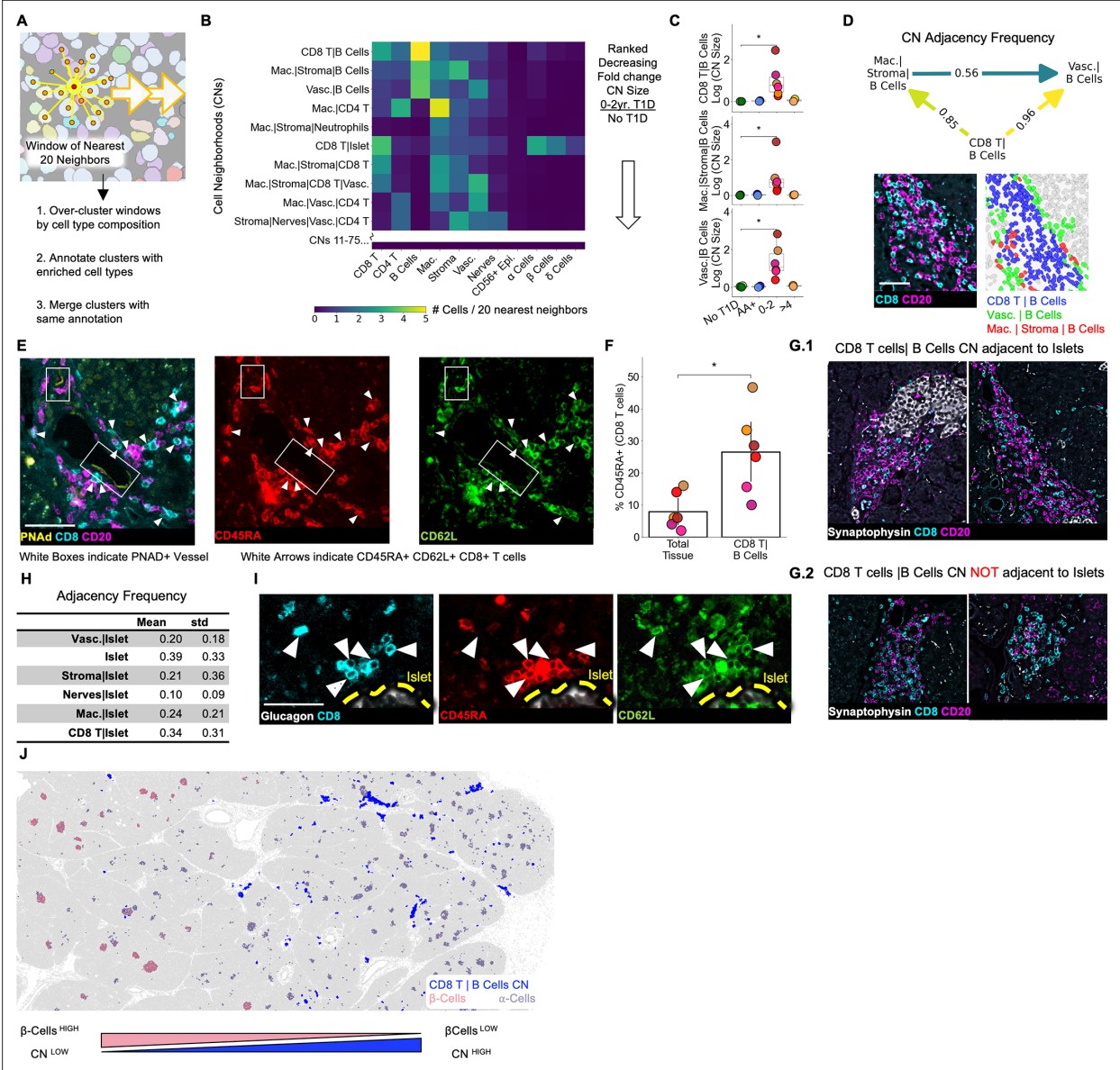

**Figure 5.** Immature tertiary lymphoid structures far from islets are potential staging areas for islet-destined CD8⁺T cells. (**A**) Schematic of algorithm for identifying CNs. Red point indicates index cell for the CN. Orange points indicate the nearest neighbors of the index cell. Windows are collected for each cell in the dataset (indicated by orange arrows). (**B**) Cell-type compositions of the top CNs organized in decreasing order of the fold increase in abundance in T1D vs. non-T1D samples. Each column in the heatmap indicates the mean density of that cell type in the 20 nearest spatial neighbors of cells assigned to the CN designated for that row. CN abundance was calculated as the number of cells assigned to the given CN divided by the number of acinar cells. Abbreviations: Vasc.: vasculature; Mac.: macrophage/DCs; Lym.: lymphatics. Neu.: neutrophils; CD8 T: CD8⁺T cells; CD4 T: CD4⁺T cells. Endocrine cell types were merged during CN annotation and are labeled "Islet". (**C**) Mean abundances of the CD8⁺T cell and B cell CNs per donor. Abundance was calculated as the number of cells assigned to the given CN divided by the number of acinar cells. Significance was determined using the Mann-Whitney U test (* p<0.05,** p<0.01, *** p<0.001). No adjustment for multiple hypothesis testing was applied. (**D**) Top: Adjacency frequencies of (CD8⁺T cells| B cell CN) with (Macrophage|Stroma|B cells) and (Vasculature| B cells) CNs. The adjacency frequency was calculated as the number of instances of the source CN adjacent to the destination CN divided by the total number of instances of the source CN. Bottom Left: Raw image of a representative assembly of the three CNs (CD8⁺T cells| B cell CN), (Macrophage|Stroma|B cells), and (Vasculature| B cells) displaying CD8 and CD20 staining. Bottom Right: The same assembly as to the left colored by CN. Scale bar indicates 50 μm. (**E**) Representative images of co-localization of PNAd⁺ endothelium and CD45RA⁺/CD62L⁺/CD8⁺T cells located in the (CD8⁺T cells|B cells) CN. Scale bar indicates 50 μm. (**F**) Frequency of CD45RA expression on CD8⁺T cells located in (CD8⁺T cell | B cell) CN relative to CD8⁺T cells throughout the tissue. Significance was determined with a Wilcoxon signed-rank test (* p<0.05, ** p<0.01, *** p<0.001). (**G**) Representative images of (CD8⁺T cells|B cells) instances adjacent to islets (G.1) and not adjacent to islets (G.2). Scale bars indicate 200 μm. (**H**) Quantification of the adjacency frequencies between the (CD8⁺T cells|B cells) CN and CNs rich in endocrine cells (i.e. 'Islet CNs'). Mean, std: mean and standard deviation adjacency frequency across T1D donors. Abbreviations: Vasc.: vasculature;

*Figure 5 continued on next page*

*Figure 5 continued*

Mac.: macrophage. (**I**) Representative images of islet-adjacent CD45RA⁺/CD62L⁺/CD8⁺T cells. Scale bar indicates 50 μm. (**J**) Image showing the spatial distribution of the (CD8⁺T cells|B cells) CN instances relative to islets and the enrichment of (CD8⁺T cells|B cells) CN instances in areas of the pancreas with islets lacking β-cells.

The online version of this article includes the following figure supplement(s) for figure 5:

**Figure supplement 1.** Association of the cell composition of the CD8⁺T cells|B cells CN with islet proximity.

**Figure supplement 2.** Correlation of key islet features with diabetes duration.

**Figure supplement 3.** Correlation of key islet features with age of onset.

other vessels (data not shown). Although we could not assess the presence of other TLS traits such as follicular dendritic cells, fibroblastic reticular cells, or follicular helper T cells, the aggregation of B cells and presence of HEVs, but the lack of compartmentalized B and T cell zones indicate that instances of the (CD8⁺T cells|B cells) CN are immature TLSs.

Next, we asked if immature TLSs could support the entry of naive CD8⁺T cells into the pancreas. We observed CD8⁺T cells co-expressing CD45RA and CD62L (the ligand for PNAd) near PNAd⁺ vasculature (*Figure 5E*, middle and right image respectively). Thus, CD45RA⁺/CD8⁺T cells in the pancreas may adhere to HEVs. Furthermore, CD45RA⁺ was enriched threefold on CD8⁺T cells in the (CD8⁺T cells|B cells) CN relative to CD8⁺T cells in the tissue as a whole (*Figure 5F*), providing additional evidence that CD45RA⁺ T cells may enter the pancreas through HEVs.

We found immature TLSs both adjacent (*Figure 5G*.1) or not adjacent (*Figure 5G*.2) to islets. Quantifying the frequency of this adjacency revealed that fewer than half of the immature TLSs were adjacent to islets (*Figure 5H*). We did not identify any differences in the cellular composition of the immature TLSs that were or were not adjacent to islets (*Figure 5—figure supplement 1*). We reasoned that even if immature TLSs were far from islets, extravasating cells may migrate to islets. Accordingly, islet-adjacent CD45RA⁺/CD8⁺T cells (that were not in islet-adjacent TLSs) co-expressed CD62L, suggesting that they originated from the (CD8⁺T cells|B cell) CN (HH). Consistent with this, in one notable tissue donor, regions of the pancreas with Insulin-Depleted islets were enriched in the (CD8⁺T cells|B cell) CN relative to regions of the pancreas with β-cell containing islets (*Figure 5J*). This spatial correlation between the (CD8⁺T cells|B cells) CN and the destruction of islets implicates immature TLSs with islet pathology even if they are not adjacent to islets (*Figure 5J*).

## Discussion

We have performed CODEX imaging and comprehensive computational analysis of whole cadaveric pancreata from T1D subjects. Our data support several conclusions.

First, our results are consistent with the model previously proposed by Damond et al, who proposed a single trajectory for insulitis, characterized by an enrichment in HLA-ABC expression, CD8⁺T cells, and macrophage/DCs (*Damond et al., 2019*).

Second, we are the first to report that IDO⁺ vasculature is present in inflamed islets but not in normal islets or islets that have lost insulin-expression (*Figure 2G*, *Figure 2H*). Furthermore, islets with IDO⁺ vasculature contained higher frequencies of CD8⁺T cells and higher expression of HLA-ABC, but not higher frequencies of macrophage/DCs compared to inflamed islets that did not contain IDO⁺ vasculature, suggesting that IDO is induced by a cytokine produced by infiltrating CD8⁺T cells such as IFN-γ (*Figure 2I*). Given IDO's established tolerogenic role, these data suggest that the loss of IDO on vasculature could be a prerequisite for β-cell death. Leveraging this checkpoint to protect transplanted β-cells from rejection has shown promise (*Alexander et al., 2002*) and could be combined with similar approaches using programmed death-ligand 1 (*Yoshihara et al., 2020*; *Castro-Gutierrez et al., 2021*).

We did not observe IDO expression on β-cells, in contrast to Anquetil et al. that report this in healthy patients (*Anquetil et al., 2018*). Our observations are consistent with data demonstrating that IDO needs to be induced. First, it has been shown with western blot and RT-PCR that human islets require cytokine stimulation to express IDO (*Sarkar et al., 2007*). Second, the human protein atlas has tested multiple IDO antibodies and demonstrated that IDO is negative in human islets via

immunohistochemistry: (https://www.proteinatlas.org). In addition, the human protein atlas's single-cell RNAseq atlas only report IDO transcripts in immune cells in healthy pancreas.

Third, we performed the first high-dimensional spatial phenotyping of CD8⁺T cells in T1D islets. We found that most T cells were antigen experienced. A small population expressed CD45RA and CD69, which could be naive, TEMRA, or Tscm cells (P). Another population expressed LAG-3, Granzyme-B, and ICOS. It is notable that only a small population of islets had Granzyme-B-expressing T cells. This could indicate that alternative mechanisms are contributing to the elimination of β-cells.

Fourth, the insulitis trajectory is comprised of four sub-clusters, each characterized by the activation profile of the islet-infiltrating CD8⁺T cells (*Figure 3C*). Multiple of these inflamed sub-clusters were present in all T1D donors (*Figure 3E*). One potential explanation for this observation is that sub-clusters are capable of inter-converting. The factors that regulate the conversion of a given islet between sub-clusters could correspond to immunoregulatory checkpoints that are critical to the progression of T1D. Unfortunately, our search for such features failed to generate any candidates (*Figure 3—figure supplement 4*). By phenotyping T cells at different distances from the islet edge, we were able to determine that the T cell activation profiles characterizing each sub-cluster were only present in the islet, not in the surrounding tissue (*Figure 3—figure supplement 3*). Unfortunately, from our data, we cannot speculate whether the insulitis sub-clusters arose due to differential stimulation of T cells that had already entered islets, differential recruitment of pre-activated T cells, or both.

Fifth, pancreatic lobules affected by insulitis are characterized by distinct tissue markers. We discovered that lobules enriched in β-cell-depleted islets were also enriched in nerves, vasculature, and Granzyme-B⁺/CD3⁻ cells, suggesting these factors may make lobules permissive to disease (*Figure 4C*). The role of islet enervation in T1D has been studied but such work has focused on nerves in the islet rather than on nerves in the exocrine tissue (*Christoffersson et al., 2020*). The Granzyme-B⁺/CD3⁻ cells could be natural killer cells; if so, they are most likely of the CD56^dim subset as CD56 was not detected on these cells. It is noteworthy that the cell types linked with direct islet invasion were distinct from those linked to lobule patterning even though both sets of cell types were found across islet and extra-islet regions. Therefore, for insulitis to consume every islet, crosstalk may be required between the cell types in the islet and extra-islet compartments. Conversely, inhibiting this interaction might contain pathology to isolated lesions.

Finally, we identify immature TLSs away from islets where CD45RA⁺/CD8⁺T cells aggregate. We also observed an inflamed islet-subcluster characterized by an abundance of CD45RA⁺/CD8⁺ T cells. It will be important to determine whether the CD45RA⁺ T cells localized around islets may have originated from immature TLSs. In mice, blocking immune egress from lymph nodes reduced the size of TLSs and halted diabetes (*Penaranda et al., 2010*). Thus, therapeutic targeting of immune cell trafficking to TLSs could help mitigate autoimmunity in human T1D.

Together, these data illuminate relationships between insulitis, the local islet microenvironment and inflammation at distal sites.

A major limitation for the study is the cohort size. Cases with documented insulitis are very rare, significantly limiting the feasibility of curating large cohorts (*Campbell-Thompson et al., 2016*). Due to this limitation, factors such as the donors' histories of drug use, durations of stay in the intensive care unit, and BMIs could not be balanced or statistically adjusted for but should be considered as they may affect exocrine inflammation but not the prevalence of insulitis (*Bruggeman et al., 2021*; *In't Veld et al., 2010*). In addition, only one of the AA+ cases, Case 6267, had detectable insulitis (*Figure 2C*). The other case, 6314 tested positive for only one autoantibody, GADA, and therefore had a significantly lower probability of progressing to overt T1D (*Ziegler et al., 2013*). Age and diabetes duration are also important criteria to consider when interpreting key results (*Figure 5—figure supplement 2* and *Figure 5—figure supplement 3*).

Another limitation is our limited perspective on myeloid cell populations. Although antibodies in our panel detect numerous myeloid markers, we failed to identify any heterogeneity in myeloid populations during insulitis. This was likely due in part to the difficulty of segmenting myeloid cells and quantifying marker expression due to their morphology. Spatial transcriptomics could be used in future studies to better define the myeloid populations and inform additions to future CODEX panels.

Lastly, our samples are two-dimensional sections which could affect some of the adjacency relations.

In conclusion, using a data-driven approach, we mapped conserved sub-states of insulitis and integrated the spatial pathology of islet and extra-islet regions into a single model of T1D pathogenesis.

The tools and computational pipelines developed here will enable further investigation of immune pathology at the tissue scale that may lead to development of therapies for T1D.

# Materials and methods

## Key resources table

| Reagent type (species) or resource | Designation | Source or reference | Identifiers | Additional information |
|---|---|---|---|---|
| Antibody | CollV | Abcam | AB_305584 | polyclonal |
| Antibody | Ki67 | BD | AB_396287 | B56 |
| Antibody | Chromogranin A | Novus | AB_3290980 | LK2H10+PHE5+CGA/414 |
| Antibody | Proinsulin | Thermo | AB_558517 | 3A1 |
| Antibody | Glucagon | Abcam | AB_297642 | K7bB10 |
| Antibody | CD8 | Santa Cruz | AB_1120718 | C8/144B |
| Antibody | CD15 | BD | AB_397181 | MMA |
| Antibody | MPO | R&D | AB_2250866 | polyclonal |
| Antibody | S100A6 | Novus | AB_10000990 | 7D11 |
| Antibody | MUC-1 | NSJ Bioreagents | AB_2864392 | 955 |
| Antibody | Cytokeratin | Biolegend | AB_439775 | C11 |
| Antibody | alphaSma | abcam | AB_2223021 | polyclonal |
| Antibody | CD57 | Biolegend | AB_2562403 | HCD57 |
| Antibody | CD44 | Biolegend | AB_312953 | IM-7 |
| Antibody | TCR g/d | Santa Cruz | AB_1130061 | H-41 |
| Antibody | NaKATPase | Abcam | AB_2890241 | EP1845Y |
| Antibody | BCL-2 | Cell Marque | AB_2864404 | 124 |
| Antibody | Galectin-3 | Thermo | AB_2136775 | A3A12 |
| Antibody | Podoplanin | Biolegend | AB_1595511 | NC-08 |
| Antibody | CD31 | Novus Bio | AB_2864381 | C31.3+C31.7+C31.10 |
| Antibody | CD45RA | Biolegend | AB_1946436 | HI100 |
| Antibody | CD69 | Novus | AB_355231 | polyclonal (AF2359) |
| Antibody | CD20 | Novus | AB_2864380 | rIGEL/773 |
| Antibody | CD16 | CST | AB_3280014 | D1N9L |
| Antibody | CD163 | Novus | AB_714951 | EDHu-1 |
| Antibody | Somatostatin | Novus | AB_2890053 | 7G5 |
| Antibody | CD206 | R&D | AB_2063019 | poly |
| Antibody | CD45 | Novus | AB_2864384 | 2B11+PD7/26 |
| Antibody | Synaptophysin | Novus | AB_10010435 | 7H12 |
| Antibody | HLA-DR | abcam | AB_2864390 | EPR3692 |
| Antibody | VISTA | CST | AB_3683060 | D1L2G |
| Antibody | IDO | CST | AB_3683091 | D5J4E |
| Antibody | biotinylated Hyaluronan Binding Protein (HABP) | Bollyky Lab Stanford University | NA | *Clark et al., 2011* |
| Antibody | HLA-ABC | BD | AB_2739161 | EMR8-5 |

*Continued on next page*

*Continued*

| Reagent type (species) or resource | Designation | Source or reference | Identifiers | Additional information |
|---|---|---|---|---|
| Antibody | TOX | CST | AB_3675995 | E6I3Q |
| Antibody | FOXP3 | Invitrogen | AB_467555 | 236 A/E7 |
| Antibody | Insulin | Sigma | AB_260137 | K36AC10 |
| Antibody | Lag3 | CST | AB_2943248 | D2G4O |
| Antibody | PD-1 | CST | AB_3675993 | D4W2J |
| Antibody | PD-L1 | CST | AB_2922774 | E1L3N |
| Antibody | CD3 | CST | AB_2922776 | D7A6E |
| Antibody | CD4 | Abcam | AB_2864377 | EPR6855 |
| Antibody | CD11c | AbCam | AB_2864379 | EP1347Y |
| Antibody | CD56 | Cell Marque | AB_3082973 | MRQ-42 |
| Antibody | CD45RO | Santa Cruz | AB_627083 | UCH-L1 |
| Antibody | ICOS | CST | AB_3676096 | D1K2T |
| Antibody | Granzyme B | Abcam | AB_2910576 | EPR20129-217 |
| Antibody | OX40 | Biolegend | AB_10639951 | Ber-ACT35 |
| Antibody | CD138 | Invitrogen | AB_11153181 | B-A38 |
| Antibody | CD68 | CST | AB_2920587 | D4B9C |
| Antibody | PNAD | Biolegend | AB_493554 | MECA-79 |
| Antibody | CD62L | SCBT | AB_3683092 | B-8 |
| Antibody | mouse IgG | Sigma | AB_1163670 | |
| Antibody | rat IgG | Sigma | AB_1163627 | |
| Sequence-based reagent | | TriLink Biotechnologies and Integrated DNA Technologies | | *Schürch et al., 2020* |
| Biological sample (*Homo sapiens*) | FFPE tissue block | Network of Pancreatic Organ Donors | | |
| Peptide, recombinant protein | Streptavidin-PE | Biolegend | 405203 | |
| Chemical compound, drug | CODEX Reagents | | | *Schürch et al., 2020* |
| Software, algorithm | CODEX Toolkit | https://github.com/nolanlab/CODEX; *Samusik et al., 2018* | | |
| Software, algorithm | ImageJ (Fiji version 2.0.0) | https://imagej.net | | |
| Software, algorithm | VGG Image annotator | https://www.robots.ox.ac.uk/~vgg/software/via/via_demo.html | | *Dutta and Zisserman, 2019* |
| Software, algorithm | CellSeg | https://michaellee1.github.io/CellSegSite/ | | *Lee et al., 2022* |

## Human tissues

Cadaveric pancreatic FFPE tissue sections were obtained through the nPOD program, sponsored by the Juvenile Diabetes Research Fund. Case numbers cited herein are assigned by nPOD and comparable across nPOD-supported projects. 17 cases in the nPOD biorepository had been previously documented to contain insulitis. For each of these cases, we examined the triple stained immunohistochemistry images (CD3, Insulin, and Glucagon) using nPOD's online pathology database to select blocks in which insulitis was present. To ensure that the tissue regions still contained insulitis (and had not been sectioned extensively after their images were uploaded to the nPOD pathology

database), we re-sectioned and visualized CD3, Insulin, and Glucagon. The use of cadaveric human tissue samples is not considered human subject research and does not require review by Stanford University's Institutional Review Board.

## CODEX data collection
### CODEX antibody generation and validation

Oligonucleotides were conjugated to purified, carrier-free, commercially available antibodies as previously described (*Schürch et al., 2020*; *Kennedy-Darling et al., 2021*). For validation experiments, human tonsils and non-diabetic pancreata were co-embedded in FFPE blocks so both tissues could be stained and imaged simultaneously. Each antibody in the CODEX panel was validated by co-staining with previously established antibodies targeting positive and negative control cell types. Once validated, the concentration and imaging exposure time of each antibody were optimized. The tissue staining patterning was compared to the online database, The Human Protein Atlas, and the published literature. The specificity, sensitivity, and reproducibility of CODEX staining has been previously validated (*Schürch et al., 2020*; *Phillips et al., 2021a*; *Kennedy-Darling et al., 2021*; *Black et al., 2021*; *Phillips et al., 2021b*).

### CODEX staining

Staining and imaging was conducted as previously described (*Schürch et al., 2020*; *Kennedy-Darling et al., 2021*; *Black et al., 2021*; *Phillips et al., 2021b*). Briefly, FFPE tissues were deparaffinized and rehydrated. Heat-induced epitope retrieval (HIER) antigen retrieval was conducted in Tris/EDTA buffer at pH9 (Dako) at 97°C for 10 min. Tissues were blocked for 1 hr with rat and mouse Ig, salmon-sperm

**Table 3.** Standard CODEX experimental details.

| Cycle | Antibody | A488 | Exposure (ms) | Antibody | A555 | Exposure (ms) | Antibody | A647 | Exposure (ms) |
|---|---|---|---|---|---|---|---|---|---|
| 1 | CollIV | 33 | 333 | NaKATPase | 36 | 100 | HLA-ABC | 53 | 100 |
| 2 | blank | | 1000 | blank | | 1000 | blank | | 1000 |
| 3 | Ki67 | 6 | 100 | BCL-2 | 46 | 500 | TOX | 28 | 150 |
| 4 | Chromogranin A | 43 | 16 | empty | | 1 | FOXP3 | 61 | 1000 |
| 5 | Proinsulin | 63 | 40 | Galectin-3 | 60 | 166 | empty | | 1 |
| 6 | Glucagon | 24 | 50 | Podoplanin | 32 | 500 | Insulin | 30 | 200 |
| 7 | CD8 | 8 | 125 | CD31 | 68 | 100 | Lag3 | 42 | 500 |
| 8 | CD15 | 14 | 40 | CD45RA | 7 | 333 | PD-1 | 23 | 500 |
| 9 | MPO | 51 | 117 | CD69 | 52 | 500 | PD-L1 | 11 | 500 |
| 10 | S100A6 | 70 | 500 | empty | | 1 | CD3 | 77 | 500 |
| 11 | MUC-1 | 21 | 33 | CD20 | 48 | 167 | CD4 | 20 | 500 |
| 12 | Cytokeratin | 67 | 100 | CD16 | 15 | 250 | CD11c | 49 | 500 |
| 13 | alphaSma | 69 | 50 | CD163 | 45 | 100 | empty | | 1 |
| 14 | CD57 | 57 | 300 | Somatostatin | 2 | 100 | CD56 | 29 | 333 |
| 15 | CD44 | 44 | 250 | CD206 | 55 | 400 | CD45RO | 3 | 500 |
| 16 | TCR g/d | 72 | 1000 | CD45 | 56 | 250 | ICOS | 41 | 500 |
| 17 | empty | | 1 | Synaptophysin | 26 | 250 | Granzyme B | 81 | 100 |
| 18 | empty | | 1 | HLA-DR | 65 | 250 | OX40 | 66 | 400 |
| 19 | empty | | 1 | VISTA | 79 | 500 | CD138 | 76 | 200 |
| 20 | empty | | 1 | IDO | 59 | 2500 | CD68 | 5 | 100 |
| 21 | empty | | 1 | HABP | StrPE | 13 | Draq 5 | | 115 |

**Table 4.** Immature TLS CODEX experimental details.

| Cycle | Antibody | A488 | Exposure (ms) | Antibody | A555 | Exposure (ms) | Antibody | A647 | Exposure (ms) |
|---|---|---|---|---|---|---|---|---|---|
| 1 | Chromogranin A | 43 | 16 | CD20 | 48 | 167 | CD3 | 77 | 500 |
| 2 | Proinsulin | 63 | 40 | CD31 | 68 | 100 | CD45RO | 3 | 500 |
| 3 | CD8 | 8 | 125 | CD45RA | 7 | 333 | CD4 | 20 | 500 |
| 4 | Ki67 | 6 | 100 | Podoplanin | 32 | 500 | CD138 | 76 | 200 |
| 5 | Glucagon | 24 | 50 | CD62L | 38 | 250 | PnAD | 71 | 333 |
| 6 | alphaSma | 69 | 50 | | | | CollIV | 33 | 333 |

DNA, and a mixture of the non-fluorescent DNA oligo sequences used as CODEX barcodes. Tissues were stained with the antibody cocktail in a sealed humidity chamber overnight at 4°C with shaking. The next day, tissues were washed, fixed with 1.6% paraformaldehyde, 100% methanol, and BS3 (Thermo Fisher Scientific), and mounted to a custom-made acrylic plate attached to the microscope.

## CODEX imaging

Imaging was conducted using the Keyence BZ-X710 fluorescence microscope with a CFI Plan Apo $\lambda$ 20x/0.75 objective (Nikon). 'High resolution' mode was selected in Keyence Navigator software, resulting in a final resolution of .37744 µm/pixel. The exposure times are listed in *Tables 3 and 4*. Regions for imaging were selected by rendering HLA-ABC, Proinsulin, and CD8 and selecting large regions (averaging 55 mm²). The full antibody panel and cycle ordering is detailed in *Tables 3 and 4*. Biotinylated hyaluronan-binding protein *Clark et al., 2011* was rendered by adding streptavidin-PE at 1:500 concentration to the 96-well plate containing fluorescent oligos in the last cycle and running the CODEX program normally. DRAQ5 was added to the last cycle because we found it stained nuclei more evenly than HOECHST which slightly improved segmentation. Each tissue took between 3 and 7 days to image depending on the tissue area.

## Characterization of tertiary lymphoid structures

A serial section from case 6228 was imaged in a separate CODEX experiment using an antibody panel tailored for characterizing tertiary lymphoid structures, as described in *Table 4*. These data were acquired and analyzed identically to the main dataset.

## Image pre-processing

Drift compensation, deconvolution, z-plane selection was performed using the CODEX Toolkit uploader (https://github.com/nolanlab/CODEX; *Goltsev et al., 2018*). Cell segmentation using the DRAQ5 nuclear channel and lateral bleed compensation was performed with CellSeg (*Lee et al., 2022*). Raw and processed data were deposited in the BioStudies BioImage Archive with the accession number S-BIAD859 (*Bollyky and Graham, 2023*; *Sarkans et al., 2018*).

## **Cell type clustering and annotation**

The source code and intermediate files for the following analyses were deposited with the imaging data in the BioStudies BioImage Archive with the accession number S-BIAD859 (*Bollyky and Graham, 2023*; *Sarkans et al., 2018*).

Marker expression was z-normalized within each donor and subsequently clustered in two steps. First, cells were projected into two dimensions using the markers indicated in *Table 2* and Parametric Uniform Manifold Approximation and Projection (pUMAP; *Sainburg et al., 2021*) was applied on a downsampled dataset. The fit model was used to transform the remaining cells. Cell types were gated using Leiden clustering and manual merging. The cluster containing immune cells was sub-clustered using the markers detailed in *Table 2*. Acinar cells contaminating the Immune cluster were gated out and merged with the Acinar cluster from the previous step. The Endocrine class was sub-clustered into α-, β-, and δ-Cells using Glucagon, Proinsulin, and Somatostatin, respectively. Clusters were

annotated according the heatmap marker expression, and overlaying annotations onto raw images (*Figure 1—figure supplement 1*) using custom scripts in Fiji.

## Islet segmentation and pseudotime
### Preprocessing
Windows consisting of the twenty nearest spatial neighbors surrounding each single cell were clustered according to their cell type composition using Mini Batch K Means with k=200. For this analysis, α-, β-, and δ-Cells were combined into one 'Endocrine' cell type. One cluster was highly enriched in Endocrine cells and accurately defined the islet area. Individual islets were identified using the connected components algorithm and filtering out islets that had fewer than ten total cells.

### PAGA analysis
For each islet, the number of each cell type inside the islet and between the islet edge and 20 μm beyond were extracted. To adjust for variation due to the islet size, the cell type counts were divided by the number of endocrine cells inside the islet. Data were then log-transformed. The PAGA embedding was computed using the default parameters except for the following: The neighborhood search was performed using cosine distance and 15 nearest neighbors; Leiden clustering used a resolution of 1. For computing the pseudotime values (used in the colormap in *Figure 2B*, the x-axis in *Figure 2F* and *Figure 4*), the path through the inflamed islet was isolated by temporarily omitting 25 islets positioned in the middle of the map between Normal and Insulin-Depleted islets. Only nine of these were from T1D or AA+ donors.

## Quantification and validation of functional marker gating
### Annotation of ground-truth dataset
4000 CD8+T cells were labeled for 15 markers by an immunologist familiar with the staining patterns of each marker using VGG Image annotator (*Dutta and Zisserman, 2019*).

### Automated thresholding
For each functional marker of interest, the lateral-bleed-compensated mean fluorescence *Lee et al., 2022* of cell types known to not express the marker in question were used to calculate a background distribution. Marker-positive cells were defined as those whose expression was greater than the 99th percentile of the background distribution.

### Gating with neural network
22μm x 22μm cropped images of each single cell were used as training data. The marker that the image corresponded to was not included as an input in the neural network and one classifier was trained for all markers. Cells were split into training, validation, and test splits (60/15/25 respectively). ResNet50 architecture and initial weights were imported from the Keras library pre trained on ImageNet. Image augmentation consisted of random flips, rotations, zooms, contrast, and translation (±ten pixels only). All weights were unfrozen, and the model was trained for 100 epochs (see accompanying source code for training details).

## Sub-clustering of inflamed islets with cell-type-specific functional markers
For each Inflamed Islet (n=351), the frequency of each marker expressed by CD8+T cells was computed. Single cells inside the islet and within 20 μm from the islet's edge were combined before the frequency was measured. The subsequent matrix underwent z-normalization followed by UMAP gating using Bokeh. Insulin-Depleted + Immune Islets were defined as islets without β-cells with greater than two CD8+T cells and greater than seven macrophage/DCs. These thresholds correspond to the 95th percentiles of CD8+T cells and macrophage/DCs in Normal islets.

## Identification of cellular neighborhoods
Previously, CNs (*Schürch et al., 2020*) were identified by, for each single cell, defining its 'window' as the 20 spatial nearest neighbors. Cells were clustered according to the number of each cell type

in their windows using Mini Batch K-Means. The output clusters corresponded to CNs. To ensure our method was sensitive to rare neighborhoods, we adapted this algorithm by over-clustering, using k=200 in the K-Means step rather than using a k ranging from 10 to 20 as used elsewhere (*Phillips et al., 2021a*; *Bhate et al., 2022*; *Shekarian et al., 2022*). Next, to determine which cell types were characteristic of each cluster, we identified, for each cluster, the set of cell types that were present in more than 80% of the windows allocated to that cluster. We named the clusters according to this set of cell types and merged all clusters with the same name, resulting in 75 CNs. Acinar cells and epithelial cells were used in the kNN graph and in the clustering but were not considered when merging clusters. Note that this method does not differentiate neighborhoods that have the same combination of cell types but different stoichiometries.

### Lobule segmentation

A training dataset was generated by manually tracing the edges of lobules in ImageJ using the ROI function. The ROI were then floodfilled in Python and used as masks for training. For each tile, the blank cycle was selected to distinguish tissue from background coverslip. A U-Net model was trained for 10 epochs (see attached source code for training details). After stitching together all masks, the resulting images required slight refinement where lobules were not completely separated, and this was done manually in ImageJ. The connected components in the stitched image defined the lobule instances. Cells were assigned to a lobule by indexing the lobule mask with their X and Y coordinates. Cells in the inter-lobular space were assigned to one 'edge' lobule. This resulted in 464 lobules.

### Formulation of hierarchical linear models

For each lobule, the number of each cell type in the extra-islet space was divided by the number of acinar cells in the extra-islet space. For all HLMs, the *lme4* package for R was used *Bates et al., 2015* and statistical significance was computed using the *lmerTest* package for R (*Kuznetsova et al., 2017*). Lobular cell type abundance was z-normalized within each donor and the pseudotime was z-normalized across the entire dataset prior to fitting.

The ICC was computed using the model: $pseudotime_{islet} \sim 1|lobuleID$ with the *performance* package in R. A value of 0 indicates that the variation in pseudotimes of islets within the same lobule is equal to the variation across all islets in the donor and a value of 1 indicates that the variation in pseudotimes of islets within the same lobule is much smaller than that of all islets in the donor.

For each cell type, a two-level, random intercept HLM within each donor was constructed with the following formulation (in R formula syntax): $pseudotime_{islet} \sim celltype_{lobule} + (1|lobuleID)$ and a three-level random intercept, random slope HLM including islets from all donors was formulated: $pseudotime_{islet} \sim celltype_{lobule} + (1 + celltype_{lobule}|donorID) + (1|lobuleID)$. Here, $pseudotime_{islet}$ equals the pseudotime of each islet, $celltype_{lobule}$ equals the number of the given cell type in a particular lobule divided by the number of acinar cells in that lobule, z-normalized within each donor, and *lobuleID* and *donorID* are categorical variables specifying the lobule and donor that the given islet belongs to.

### Neighborhood adjacency

The adjacency between neighborhoods was computed as described previously (*Bhate et al., 2022*). The only modification was that neighborhood instances were identified using connected components of the k-NN graph with k=5 rather than from the thresholded images.

## Acknowledgements

The content and views expressed are the responsibility of the authors and do not necessarily reflect the official view of nPOD. Organ Procurement Organizations (OPO) partnering with nPOD to provide research resources are listed at http://www.jdrfnpod.org/for-partners/npod-partners/. The content is solely the responsibility of the authors and does not necessarily represent the official views of the National Institutes of Health. We like to thank Yury Goltsev, Pauline Chu, Sarah Black, Gustavo Vazquez, Aviv Hargil (Stanford University), and Irina Kusmartseva (nPOD) for excellent assistance. We like to thank Dr. Xavier Rovira-Clavé (Stanford University) for critical comments on the manuscript.

# Additional information

## Competing interests

Christian M Schürch: Scientific advisory board of, stock options in, research funding from Enable Medicine, Inc. Gernot Kaber: Filed intellectual property around 4-MU; holds a financial interest in Halo Biosciences, a company that is developing 4-MU for various indications. Nadine Nagy, Paul L Bollyky: Founder, Halo Biosciences; filed intellectual property around 4-MU; holds a financial interest in Halo Biosciences, a company that is developing 4-MU for various indications. Jeffrey A Bluestone: Board of director for Gilead and CEO and President of Sonoma Biotherapeutics; scientific advisory boards of Arcus Biotherapeutics and Cimeio Therapeutics; consultant for Rheos Medicines, Provention Bio; stockholder in Rheos Medicines, Vir Therapeutics, Arcus Biotherapeutics, Solid Biosciences, Celsius Therapeutics; Gilead Sciences, Provention Bio, Sonoma Biotherapeutics. Garry P Nolan: Has received research grants from Vaxart and Celgene during the course of this work and has equity in and is a scientific advisory board member of Akoya Biosciences; Akoya Biosciences makes reagents and instruments that are dependent on licenses from Stanford University; Stanford University has been granted US patent 9909167, which covers some aspects of the CODEX imaging platform used in this manuscript. The other authors declare that no competing interests exist.

## Funding

| Funder | Grant reference number | Author |
|---|---|---|
| Breakthrough T1D | nPOD RRID:SCR_014641 | Jeffrey A Bluestone |
| Breakthrough T1D | nPOD: 5-SRA-2018-557-Q-R | Jeffrey A Bluestone |
| Helmsley Charitable Trust | #2018PG-T1D053,G-2108-04793 | Jeffrey A Bluestone |
| National Institutes of Health | K99CA246061 | Christian M Schürch |
| National Institutes of Health | 5U54CA209971-05 | Jeffrey A Bluestone Garry P Nolan Paul L Bollyky |
| National Institutes of Health | 5U2CCA233195-02 | Jeffrey A Bluestone Garry P Nolan Paul L Bollyky |
| National Institutes of Health | 1U2CCA233238-01 | Jeffrey A Bluestone Garry P Nolan Paul L Bollyky |
| National Institutes of Health | 5U01AI101984-09 | Jeffrey A Bluestone Garry P Nolan Paul L Bollyky |
| Swiss National Science Foundation | P300PB_171189 | Christian M Schürch |
| Swiss National Science Foundation | P400PM_183915 | Christian M Schürch |

The funders had no role in study design, data collection and interpretation, or the decision to submit the work for publication.

## Author contributions

Graham L Barlow, Conceptualization, Data curation, Software, Formal analysis, Investigation, Methodology, Writing – original draft, Writing – review and editing; Christian M Schürch, Salil S Bhate, Darci J Phillips, Sasvath Ramachandran, Janet Meng, Eva Korpos, Methodology; Arabella Young, Shen Dong, Hunter A Martinez, Gernot Kaber, Nadine Nagy, Investigation; Jeffrey A Bluestone, Garry P Nolan, Paul L Bollyky, Conceptualization, Supervision, Funding acquisition, Writing – original draft, Writing – review and editing

## Author ORCIDs
Graham L Barlow (iD) https://orcid.org/0000-0001-9335-5414
Jeffrey A Bluestone (iD) https://orcid.org/0000-0001-8793-7848
Paul L Bollyky (iD) https://orcid.org/0000-0003-2499-9448

Reviewer #1 (Public review): https://doi.org/10.7554/eLife.100535.3.sa1
Reviewer #2 (Public review): https://doi.org/10.7554/eLife.100535.3.sa2
Reviewer #3 (Public review): https://doi.org/10.7554/eLife.100535.3.sa3
Author response https://doi.org/10.7554/eLife.100535.3.sa4

## Additional files

### Supplementary files
MDAR checklist

### Data availability
All data is hosted by the BioImage Archive (https://www.ebi.ac.uk/bioimage-archive/) with the accession number S-BIAD859. This includes the raw images, stitched, processed images, single cell dataframes, and all supplemental code used to generate the manuscript's figures. The code is completely open source. The code is not intended as a software tool and so no small example data set is applicable.

The following dataset was generated:

| Author(s) | Year | Dataset title | Dataset URL | Database and Identifier |
|---|---|---|---|---|
| Paul B, Graham B | 2023 | High-Parameter Spatial Profiling of the Pancreas in Human Type 1 Diabetes | https://www.ebi.ac.uk/biostudies/bioimages/studies/S-BIAD859 | bioimages, S-BIAD859 |

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
