## [Editor Report · eLife Assessment]

This **valuable** study leverages innovative high-dimensional imaging strategies to interrogate pancreatic immune cell profiles and distributions throughout stages of type 1 diabetes (T1D). Despite a notable limitation in the number of donor samples analyzed, the authors identify a series of intriguing "immune signatures" and histopathological features that collectively constitute a **solid** foundation for future investigations into immunological processes underpinning the pathogenesis of T1D. Accordingly, the work will be of considerable interest to the community of T1D researchers and clinicians.

---

## [Referee Report · Reviewer #1 (Public review)]

Summary:

Barlow and coauthors utilized the high-parameter imaging platform of CODEX to characterize the cellular composition of immune cells in situ from tissues obtained from organ donors with type 1 diabetes, subjects presented with autoantibodies who are at elevated risk, or non-diabetic organ donor controls. The panels used in this important study were based up prior publications using this technology, as well a priori and domain specific knowledge of the field by the investigators. Thus, there was some bias in the markers selected for analysis. The authors acknowledge that these types of experiments may be complemented moving forward with the inclusion of unbiased tissue analysis platforms that are emerging that can conduct a more comprehensive analysis of pathological signatures employing emerging technologies for both high-parameter protein imaging and spatial transcriptomics.

Strengths:

In terms of major findings, the authors provide important confirmatory observations regarding a number of autoimmune-associated signatures reported previously. The high parameter staining now increases the resolution for linking these features with specific cellular subsets using machine learning algorithms. These signatures include a robust signature indicative of IFN-driven responses that would be expected to induce a cytotoxic T cell mediated immune response within the pancreas. Notable findings include the upregulation of indolamine 2,3-dioxygenase-1 in the islet microvasculature. Furthermore, the authors provide key insights as to the cell:cell interactions within organ donors, again supporting a previously reported interaction between presumably autoreactive T and B cells.

Weaknesses:

These studies also highlight a number of molecular pathways that will require additional validation studies to more completely understand whether they are potentially causal for pathology, or rather, epiphenomenon associated with increased innate inflammation within the pancreas of T1D subjects. Given the limitations noted above, the study does present a rich and integrated dataset for analysis of enriched immune markers that can be segmented and annotated within distinct cellular networks. This enabled the authors to analyze distinct cellular subsets and phenotypes in situ, including within islets that peri-islet infiltration and/or intra-islet insulitis.

Despite the many technical challenges and unique organ donor cohort utilized, the data are still limited in terms of subject numbers - a challenge in a disease characterized by extensive heterogeneity in terms of age of onset and clinical and histopathological presentation. Therefore, these studies cannot adequately account for all of the potential covariates that may drive variability and alterations in the histopathologies observed (such as age of onset, background genetics, and organ donor conditions). In this study, the manuscript and figures could be improved in terms of clarifying how variable the observed signatures were across each individual donor, with the clear notion that non-diabetic donors will present with some similar challenges and variability.

---

## [Referee Report · Reviewer #2 (Public review)]

Summary:

The authors aimed to characterize the cellular phenotype and spatial relationship of cell types infiltrating the islets of Langerhans in human T1D using CODEX, a multiplexed examination of cellular markers

Strengths:

Major strengths of this study are the use of pancreas tissue from well-characterized tissue donors, the use of CODEX, a state-of-the-art detection technique of extensive characterization and spatial characterization of cell types and cellular interactions. The authors have achieved their aims with the identification of the heterogeneity of the CD8+ T cell populations in insulitis, the identification of a vasculature phenotype and other markers that may mark insulitis-prone islets, and characterization of tertiary lymphoid structures in the acinar tissue of the pancreas. These findings are very likely to have a positive impact on our understanding (conceptual advance) of the cellular factors involved in T1D pathogenesis which the field requires to make progress in therapeutics.

Weaknesses:

A major limitation of the study is the cohort size, which the authors directly state. However, this study provides avenues of inquiry for researchers to gain further understanding of the pathological process in human T1D.

Comments on revisions:

The authors have responded well to the 3 critiques. They have addressed my specific comments in their revised text.

I have no further comments.

---

## [Referee Report · Reviewer #3 (Public review)]

Summary:

The authors applied an innovative approach (CO-Detection by indEXing - CODEX) together with sophisticated computational analyses to image pancreas tissues from rare organ donors with type 1 diabetes. They aimed to assess key features of inflammation in both islet and extra-islet tissue areas; they report that the extra-islet space of lobules with extensive islet infiltration differs from the extra-islet space of less infiltrated areas within the same tissue section. The study also identifies four sub-states of inflamed islets characterized by the activation profiles of CD8+T cells enriched in islets relative to the surrounding tissue. Lymphoid structures are identified in the pancreas tissue away from islets, and these were enriched in CD45RA+ T cells - a population also enriched in one of the inflamed islet sub-states. Together, these data help define the coordination between islets and the extra-islet pancreas in the pathogenesis of human T1D.

Strengths:

The analysis of tissue from well-characterized organ donors, provided by the Network for the Pancreatic Organ Donor with Diabetes, adds strength to the validity of the findings.

By using their innovative imaging/computation approaches, key known features of islet autoimmunity were confirmed, providing validation of the methodology.

The detection of IDO+ vasculature in inflamed islets - but not in normal islets or islets that have lost insulin-expression links this expression to the islet inflammation, and it is a novel observation. IDO expression in the vasculature may be induced by inflammation and may lost as disease progresses, and it may provide a potential therapeutic avenue.

The high-dimensional spatial phenotyping of CD8+T cells in T1D islets confirmed that most T cells were antigen experienced. Some additional subsets were noted: a small population of T cells expressing CD45RA and CD69, possibly naive or TEMRA cells, and cells expressing LAG-3, Granzyme-B, and ICOS.

While much attention has been devoted to the study of the insulitis lesion in T1D, our current knowledge is quite limited; the description of four sub-clusters characterized by the

activation profile of the islet-infiltrating CD8+T cells is novel. Their presence in all T1D donors, indicates that the disease process is asynchronous and is not at the same stage across all islets. Although this concept is not novel, this appears to be the most advanced characterization of insulitis stages.

When examining together both the exocrine and islet areas, which is rarely done, authors report that pancreatic lobules affected by insulitis are characterized by distinct tissue markers. Their data support the concept that disease progression may require crosstalk between cells in the islet and extra-islet compartments. Lobules enriched in β-cell-depleted islets were also enriched in nerves, vasculature, and Granzyme-B+/CD3- cells, which may be natural killer cells.

Lastly, authors report that immature tertiary lymphoid structures (TLSs) exist both near and away from islets, where CD45RA+ CD8+T cells aggregate, and also observed an inflamed islet-subcluster characterized by an abundance of CD45RA+/CD8+ T cells. These TLSs may represent a point of entry for T cells and this study further supports their role in islet autoimmunity.

Weaknesses:

As the author themselves acknowledge, the major limitation is that the number of donors examined is limited as those satisfying study criteria are rare. Thus, it is not possible to examine disease heterogeneity, and the impact of age at diagnosis. Of 8 T1D donors examined, 4 would be considered newly diagnosed (less than 3 months from onset) and 4 had longer disease durations (2, 2, 5 and 6 years). It was unclear if disease duration impacted the results in this small cohort. In the introduction, the authors discuss that most of the pancreata from nPOD donors with T1D lack insulitis. This is correct, yet it is a function of time from diagnosis. Donors with shorter duration will be more likely to have insulitis. A related point is that the proportion of islets with insulitis is low even near diagnosis, Finally, only one donor was examined that while not diagnosed with T1D, was likely in the preclinical disease stage and had autoantibodies and insulitis. This is a critically important disease stage where the methodology developed by the investigators could be applied in future efforts.

While this was not the focus of this investigation, it appears that the approach was very much immune-focused and there could be value in examining islet cells in greater depth using the methodology the authors developed.

Additional comments

Overall, the authors were able to study pancreas tissues from T1D donors and perform sophisticated imaging and computational analysis that reproduce and importantly extend our understanding of inflammation in T1D. Despite the limitations associated with the small sample size, the results appear robust, and the claims are well supported.

The study expands the conceptual framework of inflammation and islet autoimmunity, especially by the definition of different clusters (stages) of insulitis and by the characterization of immune cells in and outside the islets.

Comments on revisions:

I have not felt the need to update the initial review.

However, I note that the paragraph describing the nPOD repository (lines 154-158) can be misinterpreted that insulitis is infrequent in T1D (17 of 200 donors had it) without the clarification that insulitis is present around the time of diagnosis in most patients and it subsides over time. Thus, authors are urged to clarify that the presence of insulitis and its severity are impacted by the disease stage and disease duration.

The last sentence of this paragraph, lines 164-165, although linked to the previous sentence about the cause of death in the donors, may be misconstrued in the context of this paragraph, and it is unclear what data support this statement. Please delete this sentence.

---

## [Author Response]

The following is the authors’ response to the original reviews.

**Public Reviews:**

**Reviewer #1 (Public review):**
Summary:Barlow and coauthors utilized the high-parameter imaging platform of CODEX to characterize the cellular composition of immune cells in situ from tissues obtained from organ donors with type 1 diabetes, subjects presented with autoantibodies who are at elevated risk, or non-diabetic organ donor controls. The panels used in this important study were based on prior publications using this technology, as well as a priori and domain-specific knowledge of the field by the investigators. Thus, there was some bias in the markers selected for analysis. The authors acknowledge that these types of experiments may be complemented moving forward with the inclusion of unbiased tissue analysis platforms that are emerging that can conduct a more comprehensive analysis of pathological signatures employing emerging technologies for both high-parameter protein imaging and spatial transcriptomics.Strengths:In terms of major findings, the authors provide important confirmatory observations regarding a number of autoimmune-associated signatures reported previously. The high parameter staining now increases the resolution for linking these features with specific cellular subsets using machine learning algorithms. These signatures include a robust signature indicative of IFN-driven responses that would be expected to induce a cytotoxic T-cell-mediated immune response within the pancreas. Notable findings include the upregulation of indolamine 2,3-dioxygenase-1 in the islet microvasculature. Furthermore, the authors provide key insights as to the cell:cell interactions within organ donors, again supporting a previously reported interaction between presumably autoreactive T and B cells.Weaknesses:These studies also highlight a number of molecular pathways that will require additional validation studies to more completely understand whether they are potentially causal for pathology, or rather, epiphenomenon associated with increased innate inflammation within the pancreas of T1D subjects. Given the limitations noted above, the study does present a rich and integrated dataset for analysis of enriched immune markers that can be segmented and annotated within distinct cellular networks. This enabled the authors to analyze distinct cellular subsets and phenotypes in situ, including within islets that peri-islet infiltration and/or intra-islet insulitis.Despite the many technical challenges and unique organ donor cohort utilized, the data are still limited in terms of subject numbers - a challenge in a disease characterized by extensive heterogeneity in terms of age of onset and clinical and histopathological presentation. Therefore, these studies cannot adequately account for all of the potential covariates that may drive variability and alterations in the histopathologies observed (such as age of onset, background genetics, and organ donor conditions). In this study, the manuscript and figures could be improved in terms of clarifying how variable the observed signatures were across each individual donor, with the clear notion that non-diabetic donors will present with some similar challenges and variability.

Thank you to all reviewers and editors for their thoughtful and constructive engagement with our manuscript. We agree that patient heterogeneity and the sample size limited the impact of this study. In the future, more cases with insulitis will become available and spatial technologies will become more scalable.

Given these constraints, we have made a significant effort to illustrate the individual heterogeneity of the disease by using the same color for each nPOD case ID throughout the manuscript and showing individual donors whenever feasible (e.g. Figures 1D-E, 2C, 2I, 3E, 3G, 4B-C, 5C, and 5F). For figures related to insulitis, we do not typically include non-T1D controls since they did not have any insulitis (Figure 2C). We also explicitly discuss the differences in the two autoantibody-positive, non-T1D cases: one closely resembled the T1D cases with respect to multiple features and the other more closely resembled the non-T1D, autoantibody-negative controls.

**Reviewer #2 (Public review):**
Summary:The authors aimed to characterize the cellular phenotype and spatial relationship of cell types infiltrating the islets of Langerhans in human T1D using CODEX, a multiplexed examination of cellular markersStrengths:Major strengths of this study are the use of pancreas tissue from well-characterized tissue donors, and the use of CODEX, a state-of-the-art detection technique of extensive characterization and spatial characterization of cell types and cellular interactions. The authors have achieved their aims with the identification of the heterogeneity of the CD8+ T cell populations in insulitis, the identification of a vasculature phenotype and other markers that may mark insulitis-prone islets, and the characterization of tertiary lymphoid structures in the acinar tissue of the pancreas. These findings are very likely to have a positive impact on our understanding (conceptual advance) of the cellular factors involved in T1D pathogenesis which the field requires to make progress in therapeutics.Weaknesses:A major limitation of the study is the cohort size, which the authors directly state. However, this study provides avenues of inquiry for researchers to gain further understanding of the pathological process in human T1D.

Thank you for your analysis. We point the reader to our above description of our efforts to faithfully report the patient variability despite the small sample size.

**Reviewer #3 (Public review):**
Summary:The authors applied an innovative approach (CO-Detection by indEXing - CODEX) together with sophisticated computational analyses to image pancreas tissues from rare organ donors with type 1 diabetes. They aimed to assess key features of inflammation in both islet and extra-islet tissue areas; they reported that the extra-islet space of lobules with extensive islet infiltration differs from the extra-islet space of less infiltrated areas within the same tissue section. The study also identifies four sub-states of inflamed islets characterized by the activation profiles of CD8+T cells enriched in islets relative to the surrounding tissue. Lymphoid structures are identified in the pancreas tissue away from islets, and these were enriched in CD45RA+ T cells - a population also enriched in one of the inflamed islet sub-states. Together, these data help define the coordination between islets and the extra-islet pancreas in the pathogenesis of human T1D.Strengths:The analysis of tissue from well-characterized organ donors, provided by the Network for the Pancreatic Organ Donor with Diabetes, adds strength to the validity of the findings.By using their innovative imaging/computation approaches, key known features of islet autoimmunity were confirmed, providing validation of the methodology.The detection of IDO+ vasculature in inflamed islets - but not in normal islets or islets that have lost insulin-expression links this expression to the islet inflammation, and it is a novel observation. IDO expression in the vasculature may be induced by inflammation and may be lost as disease progresses, and it may provide a potential therapeutic avenue.The high-dimensional spatial phenotyping of CD8+T cells in T1D islets confirmed that most T cells were antigen-experienced. Some additional subsets were noted: a small population of T cells expressing CD45RA and CD69, possibly naive or TEMRA cells, and cells expressing LAG-3, Granzyme-B, and ICOS.While much attention has been devoted to the study of the insulitis lesion in T1D, our current knowledge is quite limited; the description of four sub-clusters characterized by the activation profile of the islet-infiltrating CD8+T cells is novel. Their presence in all T1D donors indicates that the disease process is asynchronous and is not at the same stage across all islets. Although this concept is not novel, this appears to be the most advanced characterization of insulitis stages.When examining together both the exocrine and islet areas, which is rarely done, authors report that pancreatic lobules affected by insulitis are characterized by distinct tissue markers. Their data support the concept that disease progression may require crosstalk between cells in the islet and extra-islet compartments. Lobules enriched in β-cell-depleted islets were also enriched in nerves, vasculature, and Granzyme-B+/CD3- cells, which may be natural killer cells.Lastly, authors report that immature tertiary lymphoid structures (TLSs) exist both near and away from islets, where CD45RA+ CD8+T cells aggregate, and also observed an inflamed islet-subcluster characterized by an abundance of CD45RA+/CD8+ T cells. These TLSs may represent a point of entry for T cells and this study further supports their role in islet autoimmunity.Weaknesses:As the authors themselves acknowledge, the major limitation is that the number of donors examined is limited as those satisfying study criteria are rare. Thus, it is not possible to examine disease heterogeneity and the impact of age at diagnosis. Of 8 T1D donors examined, 4 would be considered newly diagnosed (less than 3 months from onset) and 4 had longer disease durations (2, 2, 5, and 6 years). It was unclear if disease duration impacted the results in this small cohort. In the introduction, the authors discuss that most of the pancreata from nPOD donors with T1D lack insulitis. This is correct, yet it is a function of time from diagnosis. Donors with shorter duration will be more likely to have insulitis. A related point is that the proportion of islets with insulitis is low even near diagnosis, Finally, only one donor was examined that while not diagnosed with T1D, was likely in the preclinical disease stage and had autoantibodies and insulitis. This is a critically important disease stage where the methodology developed by the investigators could be applied in future efforts.While this was not the focus of this investigation, it appears that the approach was very much immune-focused and there could be value in examining islet cells in greater depth using the methodology the authors developed.Additional comments:Overall, the authors were able to study pancreas tissues from T1D donors and perform sophisticated imaging and computational analysis that reproduce and importantly extend our understanding of inflammation in T1D. Despite the limitations associated with the small sample size, the results appear robust, and the claims well-supported.The study expands the conceptual framework of inflammation and islet autoimmunity, especially by the definition of different clusters (stages) of insulitis and by the characterization of immune cells in and outside the islets.

Thank you for your feedback. We agree that it would be very informative to expand on our analysis of autoantibody-positive cases and look at additional non-immune features.

**Recommendations for the authors:**

**Reviewer #1 (Recommendations for the authors):**
(1) Do any of the observed cellular or structural features correlate with age of onset or disease duration? While numbers of subjects are low, considering these as continuous variables may clarify some of the findings.

Thank you for the suggestion. In Supplemental Figure 5B-C, we plotted the key immune signatures from the manuscript against the diabetes duration and age of onset.

(2) The IDO is an interesting observation and has prior support in the literature. The authors speculate this may be induced as a feature of IFNg expressed by lymphocytes in the local microenvironment. Can any of these concepts be further validated by staining for transcription factors or surrogate downstream markers associated with Th1 skewing (e.g., Tbet, CXCR3, etc)?

The only other interferon-stimulated gene in our panel is HLA-ABC. We updated Supplemental Figure 2F to include HLA-ABC expression in IDO- and IDO+ islets (within the “Inflamed” group). Consistent with the hypothesis that IDO is stimulated by interferon, HLA-ABC is also significantly higher in IDO+ islets than IDO- islets. PDL1, another interferon-stimulated gene. was included in the panel but we did not detect any signal. This antibody was very weak during testing in the tonsil, so we couldn’t confidently claim that PDL1 was not expressed.

(3) The authors discuss the potential that CD45RA may be expressed in Temra populations. This could use additional clarification and a distinction from Tscm if possible.

Unfortunately, we did not have the appropriate markers to distinguish naive, TEMRA, or Tscm cells from each other. We updated the text in the discussion to include this consideration (Line 432).

(4) Supplemental Figure 5 is not informative in the current display.

Thank you, we replotted these data.

(5) Supplemental Table 1 could be expanded with additional metadata of interest, including the genetic features of the donors (e.g, class II diplotype and GRS2 values) that are published and available in the nPOD program.

Some genetic data are only available to nPOD investigators. We think it is more appropriate to request the data directly from them.

**Reviewer #2 (Recommendations for the authors):**
(1) I had only a few specific comments. I think the statement in Lines 317 and 318 is too strong. It implies that each lobe is always homogeneous for having all islets with insulitis or not having insulitis. Some lobes are certainly enriched for islets with insulitis but insulin+ islets without insulitis in some lobes in some T1D donors are seen. Please soften that statement.

We apologize for our lack of clarity. We have edited the text (line 305-309) to better articulate that organ donors fall on a spectrum. Thank you for raising this point as we think the motivation for our analysis is much clearer after these revisions.

(2) Please cite and discuss In't Veld Diabetes 20210 PMID: 20413508. While the main point of the paper is that there is beta cell replication after prolonged life support, another observation is that there is a correlation between prolonged life support and CD45+ cells in the pancreas parenchyma. This might indicate that not all immune cells in the parenchyma are T1D associated in donors with T1D.

Thank you, we have added this citation to our discussion of the importance of duration of stay in the ICU (Line 471).

(3) Can you rule out that CD46RA+/CD69+ CD8+ T cells in the islets are not TSCM?

(See above)

**Reviewer #3 (Recommendations for the authors):**
Similar studies in experimental models may afford increased opportunity to evaluate the significance of these findings and model their potential relevance for disease staging and therapeutic targeting.

We agree that the lack of experimental data limits the ability to interpret and validate the significance of our findings. We hope that our study motivates and helps inform such experiments.